# JMY powers dendritogenesis and is regulated by CaM revealing a general, critical principle in neuromorphogenesis
Maja Kühne[1], Anna-Lena Zepernick[1], Britta Qualmann [1,2] ✉, Michael Manfred Kessels [1,2] ✉ & Maryam Izadi-Seitz [1,2] ✉

Local calcium signals and formation of actin filaments help to steer and power neuronal morphology development and plasticity. Yet, responsible actin nucleators and their linkage to calcium transients largely remained elusive. Here, we identify the WH2 domain-based actin nucleator JMY as target of the calcium sensor calmodulin, reveal that JMY is critical for dendritic arbor formation and unravel that JMY's molecular mechanisms employed in dendritic arborization are depended on Arp2/3 complex interaction, Arp2/3 complex activity and functionality of JMY's WH2 domains, i.e. on JMY's abilities to promote actin filament formation. We furthermore demonstrate that $Ca^{2+}$/calmodulin association regulates the G-actin loading of JMY's first WH2 domain. Consistently, JMY's functions in neuromorphogenesis rely on proper $Ca^{2+}$/calmodulin signaling and on the first WH2 domain. These findings establish $Ca^{2+}$/calmodulin signaling as an important, more widely used, but multifaceted mechanism of tight control of actin nucleators powering dendritic branch formation—a key aspect in neuronal network development in the brain.

Changes in actin cytoskeletal architecture and modulations of actin dynamics play an important role in structural organization and functional adaptations of subcellular compartments, organelles, and entire cells and therefore require tight temporal and spatial control. As it is rate-limiting in filament formation, actin nucleation is particularly tightly controlled[1–3].

The polar and extremely arborized morphologies that neurons develop during pre- and postnatal brain development are a prerequisite for proper signal processing in neuronal networks. Neuronal cells rely on cytoskeletal processes to establish polarity and to form protrusive structures, such as axons, dendrites, and dendritic spines, and to extend and modulate these cellular structures during differentiation and development, as well as during plastic rearrangements and repair[4–7].

The formation of new dendritic branches seems to involve local $Ca^{2+}$ signals coinciding with transient F-actin formation at dendritic branch induction sites[8]. Recent work uncovered that the actin nucleator Cobl[9] is tightly controlled by the $Ca^{2+}$ sensor protein calmodulin (CaM). $Ca^{2+}$/CaM signaling was indispensable for the actin cytoskeletal functions of Cobl in neuromorphogenesis[8,10]. Yet, it remained open whether such $Ca^{2+}$/CaM control is a specialty of Cobl or is a crucial, versatile used cell biological principle in the control of Wiskott Aldrich syndrome protein homology 2 (WH2) domain-based actin nucleators.

The Junction-mediating and regulatory protein (JMY), initially discovered as regulating cofactor for the tumor suppressor p53[11,12] stands out as particularly interesting member of the new class of WH2 domain-based actin nucleators. JMY can nucleate actin directly via its three G-actin-binding WH2 domains and indirectly via associating with and activating the Arp2/3 complex[13,14]. JMY is highly expressed in the brain[15]. Yet, functions of JMY in neurons remained unknown.

Also, in non-neuronal cells, the cytoskeletal roles of JMY are not yet fully understood, as the results varied from cell line to cell line[12]. For example, JMY was indispensable for the motility of osteosarcoma-derived cell lines[13,14] but not for NIH3T3 cells[15]. Studies of oligodendrocytes differentiated from isolated progenitor cells argued that JMY rather positively influences cell extension[16], whereas studies in the Neuro2a neuroblastoma cell line introduced JMY as a negative regulator[15].

We here demonstrate by gain- and loss-of-function studies that JMY promotes the formation of the dendritic arbor of neurons. We also identify JMY as a binding partner of $Ca^{2+}$-activated CaM. Our studies reveal that the $Ca^{2+}$/CaM signaling pathway controls the actin association of JMY and unravel that the CaM interaction is crucial for the identified important role of JMY in the development of dendritic arbors.

[1]Institute of Biochemistry I, Jena University Hospital - Friedrich Schiller University Jena, Jena, Germany. [2]These authors contributed equally: Britta Qualmann, Michael Manfred Kessels, Maryam Izadi-Seitz. ✉e-mail: Britta.Qualmann@med.uni-jena.de; Michael.Kessels@med.uni-jena.de; Maryam.Izadi@med.uni-jena.de

Taken together, our study uncovers the functions of JMY in neuronal development, identifies an effective mechanism of regulation of the WH2 domain-based actin nucleator and actin-filament-promoting protein JMY, and reveals the molecular mechanism allowing the $Ca^{2+}$/CaM signaling pathway to steer JMY functions.

## Results

### The actin nucleator JMY is a target for CaM-mediated $Ca^{2+}$ signaling

We hypothesized that $Ca^{2+}$/CaM-mediated control of actin filament formation might perhaps be a general and important principle in the control of WH2 domain-based actin nucleators in early neuronal development and not merely a speciality of Cobl[8].

In order to address this hypothesis experimentally, we screened further actin nucleators for putative interactions with $Ca^{2+}$-activated CaM. We identified the actin nucleator JMY, which nucleates actin and, in addition, promotes actin nucleation by the Arp2/3 complex[13], as $Ca^{2+}$/CaM-target (Fig. 1A and Supplementary Fig. 1A, B). Further coprecipitations demonstrated that it was JMY's C-terminal part, comprising its three WH2 domains and its Arp2/3-binding CA domain, that was necessary and sufficient for binding to immobilized $Ca^{2+}$-activated CaM (Fig. 1A and Supplementary Fig. 1A, B). Concentration-dependent examinations of the interaction demonstrated that the interaction of JMY with $Ca^{2+}$-activated CaM was very strong (Supplementary Fig. 1C).

The JMY/CaM interaction was also observable in coprecipitation analyses with endogenous JMY from cell lysates. Also in these analyses, the JMY/CaM interaction was strongly $Ca^{2+}$-dependent (Fig. 1B).

We next aimed at vigorously testing the in vivo relevance of the discovery of JMY as a target for CaM. Endogenous JMY indeed specifically coimmunoprecipitated with endogenous CaM from HeLa cell lysates (Fig. 1C).

In order to next firmly exclude that the specific coprecipitation and coimmunoprecipitation of JMY with CaM were based on post-solubilization artifacts and to furthermore demonstrate the relevance of the identified JMY/CaM interaction in intact cells, we set up in vivo-recruitment analyses. Successful complex formation in these assays is visualized by an accumulation of a normally equally distributed or elsewhere localized binding partner to a bait protein localized to predefined locations in the cells. We used the outer mitochondrial membrane for this purpose and employed a mitochondrially targeted mCherry fusion protein of CaM (Mito-mCherry-CaM), which was developed previously and faces the cytoplasm[8,17]. Accumulation of JMY to Mito-mCherry-CaM-coated mitochondria but not to Mito-mCherry control mitochondrial visualized successful and specific reconstitutions of JMY/CaM complexes at defined sites in intact HEK293 cells (Fig. 1D).

### Overexpression of JMY promotes dendritic branching and outgrowth in primary hippocampal neurons

Our expression studies showed that JMY is expressed in hippocampal and cortical neurons and that endogenous JMY adopts a dendritic localization pattern in hippocampal neurons (Fig. 1E, F). Interestingly, JMY hereby did not only showed overlap with the dendritic marker MAP2 but was enriched in dendritic growth cones (Fig. 1E). This raised the exciting hypothesis that JMY may be among the cytoskeletal key players that are targeted by $Ca^{2+}$/CaM signaling during early neuronal development, controlling neuromorphogenesis[18]. Yet, the general role of JMY in neurons was unknown.

JMY was introduced as a negative regulator of neurite outgrowth in Neuro2a cells[15]. If the immortalized cell line Neuro2a would indeed mirror mechanisms of neuronal development, this could mean that, by interfacing with distinct actin nucleators, $Ca^{2+}$ signals may regulate different aspects in neuronal development. In oligodendrocytes, however, JMY seemed to play a positive role in the formation of cellular protrusions[16].

To explore whether JMY has a positive or a negative impact on neuromorphogenesis in primary neurons and to avoid any putative distorting effects by tag addition, we transfected primary rat hippocampal neurons at DIV4 with a vector driving the expression of untagged JMY (JMY*). GFP, which was translated from the same resulting mRNA using an internal ribosome entry site (IRES), was used as reporter to identify transfected neurons. Quantitative analyses of the reconstructed dendritic filaments according to immunostainings against the dendritic marker MAP2 revealed that neurons transiently transfected with JMY*/GFP for 30 h showed a significant increase in dendritic branch points, in dendritic terminal points and in the total dendritic length when compared to control neurons (Fig. 1G–J and Supplementary Fig. 1D, E). In primary hippocampal neurons, JMY thus clearly does not act as a negative regulator of cell morphology but promotes dendritic arbor development.

Sholl analyses (Supplementary Fig. 1E) showed statistically highly significant increases in dendritic complexity in particularly in proximal areas (Fig. 1K). Branch depth analyses (Supplementary Fig. 1E) demonstrated that JMY increases the complexity of the dendritic arbor, in particular by significant increases at branch depth levels 2, 3, and 4 (Fig. 1L).

In line with our conclusion that JMY has a promoting effect on cellular morphogenesis in neurons, we observed that JMY also strongly modulated the morphology of non-neuronal cells (Supplementary Fig. 1F). Quantitative analyses clearly demonstrated that F-actin–rich 3D membrane ruffles in COS-7 cells were more frequent in cells overexpressing JMY than in control cells (Supplementary Fig. 1G).

JMY thus promotes cytoskeletal processes, which, in neurons, manifest in form of an elevated dendritic arborization. The CaM interaction partner JMY thus seems to be a positive effector in the morphogenesis of primary hippocampal neurons.

### JMY is indispensable for proper dendritic branching

Overexpression analyses revealed that JMY has the potential to promote dendritogenesis. In order to address whether JMY might also be crucial for the development of the dendritic arbors leading to the formation of functional neuronal networks, we performed loss-of-function studies. JMY knockdown using RNAi#1 (leading to a significant knockdown of JMY based on immunofluorescence analyses of transfected neurons; Supplementary Fig. 2A, B) in developing primary hippocampal neurons led to decreases of dendritic branch points, of dendritic terminal points, and in total dendritic length compared to scrambled RNAi control (scr. RNAi) (Fig. 2A–C). The JMY loss-of-function phenotypes in dendritic arbor development were statistically highly significant and very severe. The numbers of dendritic branch points were reduced by ~50%, and those of terminal points were reduced by ~25% in JMY-deficient hippocampal neurons (Fig. 2A–D). Dendritic branch depth analyses demonstrated that, whereas the number of primary dendrites (branch depth level 1) was unchanged upon JMY deficiency, all higher branch depth levels were severely impaired. The decreases were highly statistically significant for all more abundant higher-order dendritic segments (levels 2, 3, and 4; Fig. 2E). Sholl analyses also clearly showed a reduced dendritic complexity and thereby an impaired dendritic arbor development upon JMY loss-of-function (Fig. 2F).

Importantly, reexpression of the RNAi#1-resistant JMY* with RNAi#1 rescued all loss-of-function phenotypes observed. Neurons transfected with the rescue tool RNAi#1/JMY* displayed values for dendritic branch points, dendritic terminal points, total dendritic length, Sholl intersections, and branch depth levels that were highly statistically different from RNAi#1 levels and, in part, even slightly exceeded those of scr. RNAi control neurons (Fig. 2A–F)—the latter is most likely due higher expression levels of the RNAi-insensitive JMY mutant (JMY*) compared to endogenous JMY (see Fig. 3). The successful rescue of all JMY loss-of-function phenotypes by reexpression of RNAi-insensitive JMY* clearly demonstrated the specificity of all dendritic JMY loss-of-function phenotypes we identified.

Experiments conducted with the second RNAi against JMY (RNAi#2) also highlighted an importance of JMY for the formation of complex dendritic arbors of neurons. Reexpression of a JMY mutant insensitive to JMY RNAi#2 led to a full rescue of all dendritic arbor defects caused by RNAi#2

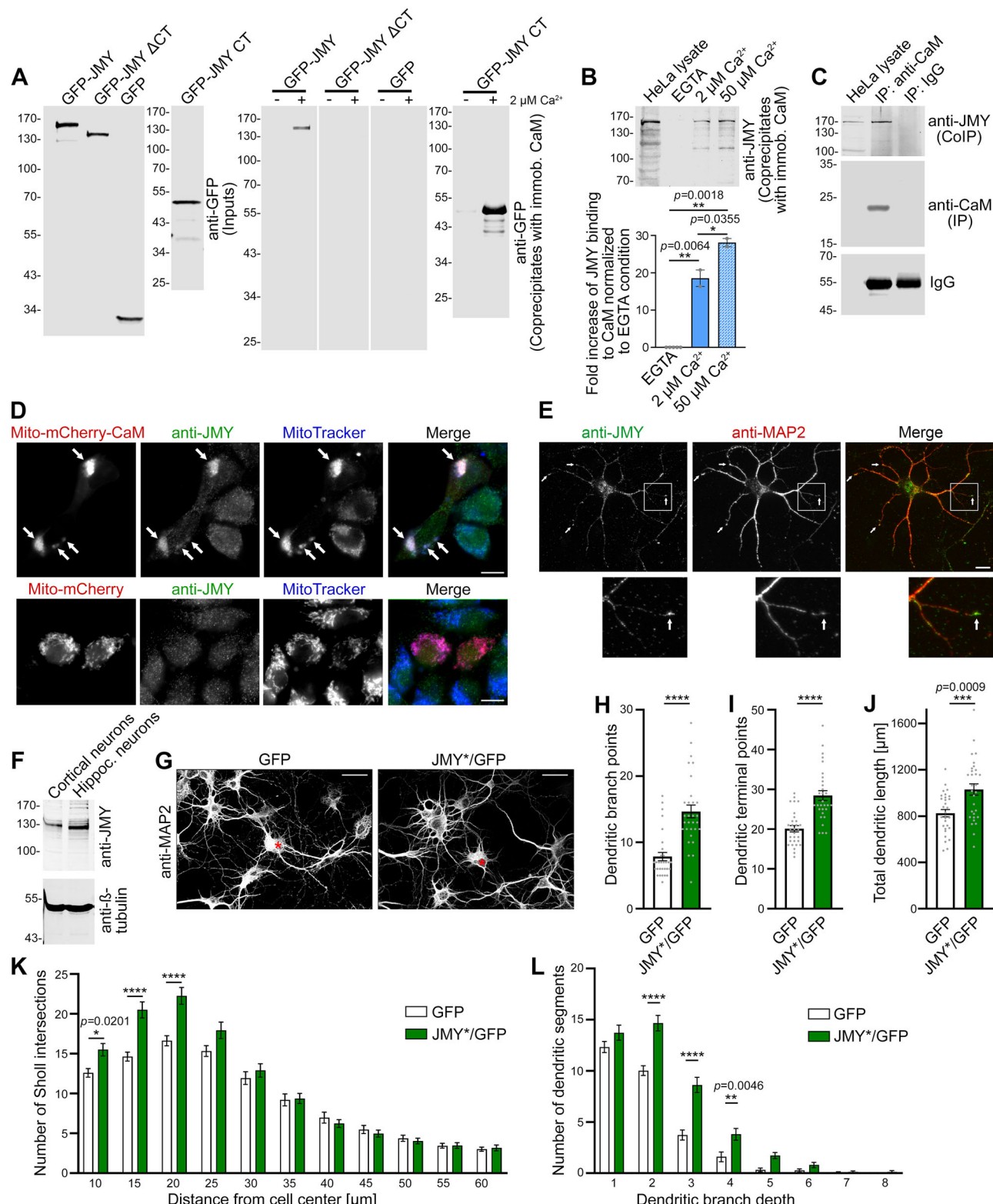

(Supplementary Fig. 2C–H). The consistency of the loss-of-function phenotypes caused by the two independent RNAi tools, as well as the successful rescue experiments for both RNAi tools, firmly proved the specificity of the identified JMY loss-of-function phenotypes.

Our functional experiments in developing primary hippocampal neurons thus clearly demonstrate a crucial role of JMY in dendritic arbor formation. Together with our gain-of-function analyses (Fig. 1), these studies demonstrate that JMY does not act as negative regulator of cell extension but instead strongly promotes neuronal morphogenesis.

## Dendritic arborisation relies on both the N- and the C-terminus of JMY

We next addressed whether the identified critical function of JMY in dendritic arborization was related to JMY's C-terminal actin binding/nucleation

**Fig. 1 | JMY is a CaM binding protein, and JMY overexpression in primary hippocampal neurons promotes dendritic branching. A** Immunoblots of coprecipitates of GFP, GFP-JMY, and fragments thereof expressed in L929 cells using a matrix with immobilized CaM and conditions without and with $Ca^{2+}$ (2 μM). **B** Immunoblot analyses of coprecipitation experiments of endogenous JMY from HeLa cell lysates with immobilized CaM and quantitative examinations thereof. Endogenous JMY interacts with CaM in a $Ca^{2+}$-dependent manner (consistently shown under two different $Ca^{2+}$ concentrations). White lines indicate lanes omitted from the blot (due to change of order). **C** Coimmunoprecipitation analyses of endogenous JMY with CaM from HeLa cells. Endogenous CaM immunoprecipitated by anti-CaM antibodies specifically coimmuniprecipitated endogenous JMY, as detected by immunobloting analyses. **D** Representative maximum intensity projections (MIPs) of HEK293 cells showing a reconstitution of JMY/CaM complexes. Note the recruitment of endogenous JMY (green in merges) to mitochondrially anchored mCherry-CaM (Mito-mCherry-CaM) but not to Mito-mCherry (red in merges). Mitochondria were marked with MitoTracker (blue in merges). Complex formation thus appears in white in the merged images (arrows mark examples). Bars, 10 μm. **E** MIPs of anti-JMY (green in merge) and anti-MAP2 (red in merge) coimmunolabeling in rat hippocampal neurons (DIV6). Growth cones displaying JMY enrichments are marked by arrows. Boxed areas represent areas magnified below. Bar, 10 μm. **F** Anti–JMY immunoblotting of lysates (30 μg protein per lane) from cultures of cortical and hippocampal rat neurons. Anti-β-tubulin immunoblotting is shown for comparison. **G** Representative MIPs of GFP and JMY*/GFP overexpressing primary hippocampal neurons immunostained with anti-MAP2 antibodies. Transfected neurons are marked by red asterisks. Bars, 30 μm. **H–L** Quantitative morphometric analyses of GFP and JMY*/GFP overexpressing primary hippocampal neurons transfected at DIV4 and fixed 30 h later. Untagged JMY* and GFP (reporter) were co-expressed as separate proteins from the same mRNA using an internal ribosome entry site (IRES). Anti-MAP2-based morphology tracing and reconstruction revealed a gain-of-function phenotype for JMY*/GFP expressing cells with significantly increased dendritic branch points (**H**), dendritic terminal points (**I**), total dendritic length (**J**), Sholl intersections (**K**) and dendritic branch depth levels 2-4 (**L**) when compared to only GFP expressing control neurons. Data, mean ± SEM shown as bar/dot plots (**B, H–J**) and bar plots (**K, L**), respectively. **B** $n = 2$ assays. **H–L** $n = 30$ cells of each condition from 2 independent assays. Numerical data and images of uncropped and unedited blots are provided in Supplementary Data 1 and 2, respectively. Statistical significances, One-way ANOVA/Tukey's post-test (**B**), Mann–Witney $U$-test (**H, I**), Welch's $t$-test (**J**), and Two-way ANOVA/Bonferroni's test (**K, L**), respectively. $*p < 0.05$, $**p < 0.01$, $***p < 0.001$, $****p < 0.0001$. For exact $p$ values see figure panels. Note that $****p < 0.0001$ are too small values to be reported by the software used.

and/or Arp2/3 complex coupling[13,15,19] or whether N-terminal functions of JMY, such as the proposed actin binding via the N-terminal part of JMY, which by some yet to be clarified mechanism seems to support JMY's actin nucleation activity[20], may be responsible for JMY's functions in dendritic arbor formation. For this purpose, we generated and analyzed two RNAi-insensitive deletion mutants of JMY (Fig. 3A). JMY* ΔCT (aa1-830), lacking all three WH2 domains and the Arp2/3 complex-binding CA domain, completely failed to rescue any of the JMY loss-of-function phenotypes in dendritic arborization. Resupplying the developing neurons with JMY* ΔCT instead of the full-length protein led to dendritic branch points, terminal points, and a total dendritic length that were indistinguishable from JMY RNAi#1 and thus remained highly statistically different from control in all parameters analyzed (Fig. 3B–E). The C-terminal G-actin- and Arp2/3 complex-binding domains of JMY domains were thus required for JMY's role in dendritogenesis. These observations were important, as recent analyses of functions of the Arp2/3 complex activator Scar/WAVE in both B16-F1 mouse melanoma and *Dictyostelium discoideum* cells demonstrated that, in contrast to our JMY results, Scar/WAVE without its VCA domain still induced the formation of morphologically normal, actin-rich protrusions, extending at comparable speeds despite a drastic reduction of Arp2/3 recruitment[21].

Investigation of cells coexpressing the corresponding JMY mutant comprising the C terminus but lacking the N terminus (RNAi#1/JMY* ΔNT) together with JMY RNAi demonstrated that the C-terminal G-actin- and Arp2/3 complex-binding domain alone was also insufficient for bringing about JMY's role in dendritic arborization. Also, expression of JMY* ΔNT (aa710-983) did not rescue the JMY loss-of-function phenotypes (Fig. 3B–E).

Sholl analyses of dendritic arbor complexity confirmed the inability of both JMY* ΔCT and JMY* ΔNT to rescue JMY deficiency phenotypes in dendritic branching (Supplementary Fig. 3) despite the fact that, similar to wild-type JMY (JMY*), both mutants were provided in excess and in soma and dendrites (Fig. 3F). Thus, both the N and the C-terminal parts of JMY are of critical importance for JMY's promoting role in dendritic arbor formation.

## JMY-mediated neuromorphogenesis critically involves the Arp2/3 complex and a direct coupling of JMY to the Arp2/3 complex

In in vitro-actin filament formation assays, JMY had been shown to produce unbranched actin filaments directly through its G-actin binding WH2 domains and to furthermore promote the nucleation of branched actin filaments via activation of the Arp2/3 complex[13]. In order to obtain first insights into a putative involvement of the Arp2/3 complex in JMY-mediated neuromorphogenesis, we used the Arp2/3 complex inhibitor CK666[22,23]. Comparisons to control neurons showed that Arp2/3 complex inhibition by CK666 specifically impaired dendritic branching and reduced dendritic branch depth, indicating that CK666-mediated arrest of the Arp2/3 complex in its inactive state has a general impact on neuronal morphology development (Fig. 4A–E). These results were somewhat in line with results of CK666 treatments in neuronal growth cones in Aplysia bag cell neurons[24].

Importantly, addition of CK666 (in DMSO) completely abolished JMY-mediated dendritic arborization (Fig. 4A–E). The values for dendritic branch points, dendritic terminal points, total dendritic length, and dendritic Sholl intersections in JMY*-expressing neurons all were statistically significantly decreased upon Arp2/3 complex inhibition with CK666. The inhibitory effects of CK666 on developing JMY*/GFP-expressing neurons were so severe that JMY*/GFP+ CK666 data resembled the GFP+ CK666 condition (Fig. 4B–E). CK666-mediated stabilization of the Arp2/3 complex in its inactive state and inhibition of its actin nucleation ability thus impaired all JMY-mediated effects on dendritic arborization.

In order to address whether the CK666 effects on JMY-mediated functions at least in part may reflect the necessity of direct Arp2/3 complex binding and activation by JMY or may instead involve theoretically also possible Arp2/3 functions downstream of JMY's own actin nucleation capability, we examined a JMY deletion mutant lacking the ability to couple to the Arp2/3 complex (Fig. 4F). Coexpression of the deletion mutant JMY* ΔCA (aa1-938) with JMY RNAi#1 against JMY was completely unable to rescue the JMY loss-of-function phenotypes in primary hippocampal neurons (Fig. 4G–I and Supplmentary Fig. 4A, B). Cells reexpressing JMY* ΔCA exhibited loss-of-function phenotypes in dendritic arborization that were comparable to the phenotypes caused by JMY RNAi#1 alone and thereby also significantly differed from control (Fig. 4G–I Supplementary Fig. 4A, B). Albeit no gross changes in actin or Arp2/3 complex localization were observable (Supplementary Fig. 4C, D), and both mutants were provided in excess and in soma and dendrites (Fig. 4J), binding to the Arp2/3 complex obviously seemed to be important for JMY-mediated functions in dendritic arbor formation.

For human Scar1/WAVE1 the C domain was reported to not only bind to the Arp2/3 complex but also to monomeric actin independently from Scar1/WAVE1's WH2 domain[25]. We therefore next excluded that deletion of the complete CA domain may not only disturb Arp2/3 complex binding but may also impair binding to G-actin or impair tertiary protein structure. An Arp2/3 complex activation-deficient JMY* point mutant (W981A[15]) yielded results very similar to the rescue attempt with JMY* ΔCA. Also, in this case, dendritic branch points, dendritic terminal points, total dendritic length, and dendritic Sholl intersections were significantly lower compared to successfully rescued neurons expressing RNAi#1/JMY* or control neurons (scr. RNAi). The failure of JMY*[W981A] to rescue the JMY loss-of-

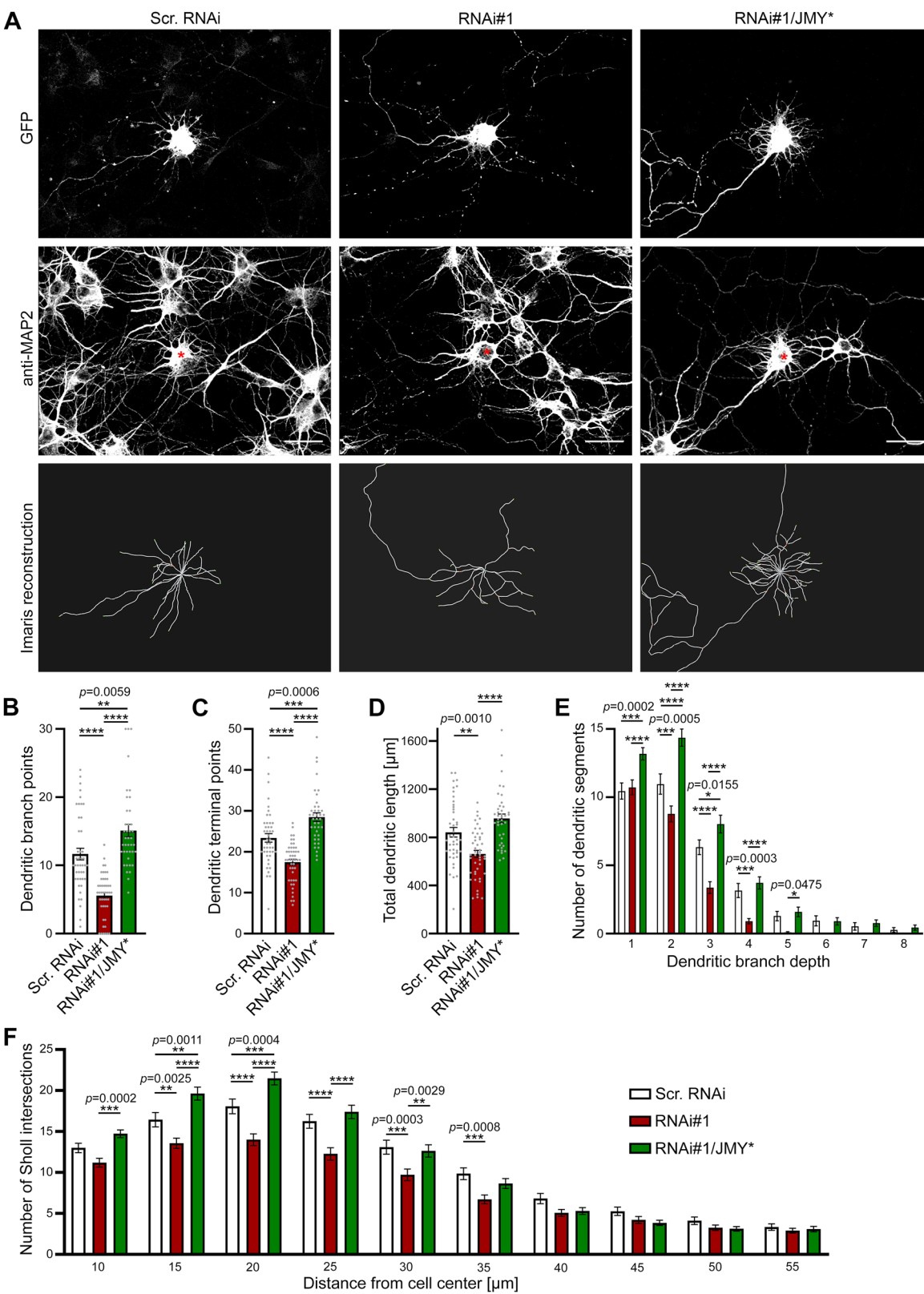

function phenotypes in dendritic branching was so severe that JMY RNAi#1/JMY*$^{W981A}$ data were indistinguishable from or even below the JMY RNAi#1 data (Fig. 4G–I and Supplementary Fig. 4A, B). Thus, the W981A point mutation completely abolished the ability of JMY* to rescue the loss-of-function phenotype.

No gross changes in actin or Arp2/3 complex localization were observable (Supplementary Fig. 4C, D) and both mutants were provided in excess and in soma and dendrites (Fig. 4J). All together, these results highlight the importance of both binding and activation of the Arp2/3 complex by JMY for proper formation of dendritic arbors.

**Fig. 2 | JMY is required for proper dendritic arborization. A** Anti-MAP staining and GPF signals of primary hippocampal neurons transfected with plasmids driving the expression of either scrambled RNAi (Scr. RNAi) or JMY RNAi (RNAi#1) together with GFP as a marker or together with GFP and an RNAi#1-insensitive JMY using an IRES (RNAi#1/JMY*) at DIV4 and fixed 30 h thereafter. Asterisks mark transfected neurons. Bars, 30 µm. Lower panels show 2D representations of 3D morphometric reconstructions by using Imaris software. Note that viewing the images at high magnification also allows for seeing the Imaris-based markings of branch points (red) and terminal points (green). Quantitative determinations of specific defects in dendritic arborization caused by JMY deficiency by addressing dendritic branch points (**B**), dendritic terminal points (**C**), total dendritic tree length (**D**), as well as by conducting dendritic branch depth (**E**) and Sholl analyses (**F**). Data, mean ± SEM shown as bar/dot plots (**B–D**) and bar plot (**E**, **F**), respectively. Scr. RNAi, $n = 44$; RNAi#1, $n = 46$; RNAi#1/JMY*, $n = 40$ cells from three independent neuronal preparations. Numerical data are provided in Supplementary Data 1. Statistical significances, One-way ANOVA/Tukey's posttest (**B–D**) and two-way ANOVA/Bonferroni's test (**E**, **F**), respectively. *$p < 0.05$, **$p < 0.01$, ***$p < 0.001$, ****$p < 0.0001$. For exact $p$ values see figure panels. Note that ****$p < 0.0001$ are too small values to be reported by the software used.

## The actin-binding ability of JMY's three WH2 domains is crucial for neuromorphogenesis

In in vitro-actin filament formation assays, JMY's G-actin-binding WH2 domains were required for JMY's WH2 domain-based actin nucleation activity and, at least in part, also for Arp2/3 complex activation[13]. Functional experiments in developing primary hippocampal neurons highlighted the importance of functional JMY WH2 domains for dendritic arborization (Fig. 5A–C and Supplementary Fig. 5A, B). Whereas reexpression of RNAi-resistant wild-type JMY (JMY*) successfully rescued all JMY loss-of-function phenotypes, rescue attempts with a JMY* mutant with all three WH2 domains carrying point mutations abolishing actin interactions (JMY*[WH2#1-3mut]) failed to elicit any beneficial effects. Dendritic branch points, terminal points, and total dendritic length in JMY RNAi/ JMY*[WH2#1-3mut]-expressing neurons did not differ from data obtained upon JMY RNAi (Fig. 5A–C), albeit the mutant was provided in excess and was available in soma and dendrites (Fig. 5D). Also, Sholl analyses showed that the WH2 domain-mediated actin binding abilities of JMY were essential for dendritic arbor formation (Supplementary Fig. 5B).

## The first WH2 domain of JMY coincides with a CaM-binding region of JMY, and this CaM-binding region is critical for JMY's functions in dendritic arbor formation

We identified JMY as a $Ca^{2+}$-dependent interaction partner of CaM (Fig. 1) and revealed that CaM associates with the C-terminal part of JMY encompassing the domains responsible for JMY's Arp2/3 complex binding and for JMY's actin binding and nucleation. In order to reveal the molecular mechanisms of a putative control of JMY functions by $Ca^{2+}$/CaM signaling, we more precisely determined the CaM binding site in JMY. Coprecipitation experiments with lysates of L929 cells expressing JMY's three individual WH2 domains in combinations with their N and C terminal linker regions as GFP-fusion proteins showed that immobilized CaM specifically associated with JMY's first WH2 domain in a wide range of $Ca^{2+}$ concentrations but not under $Ca^{2+}$-free conditions (Fig. 5E and Supplementary Fig. 5C). In contrast, CaM did not bind to the GFP fusion proteins comprising the WH2 domains #2 and #3 or their respective linker regions (Fig. 5E and Supplementary Fig. 5C).

In line, immobilized CaM specifically coprecipitated purified GST-JMY WH2#1 but not GST in a $Ca^{2+}$-dependent manner (Fig. 5F). Thus, the first WH2 domain of JMY at least partially overlaps with a CaM binding site, and the identified CaM interaction is direct.

Based on our biochemical studies delimiting the CaM binding sites in the JMY C-terminus, we established a CaM binding-deficient JMY deletion mutant (Supplementary Fig. 5D). Integrating this mutation into the context of an RNAi-insensitive full-length JMY (JMY*ΔCaM; Δ829-876) (Fig. 5G) allowed for vigorously testing the relevance of the identified CaM binding region of JMY for JMY's functions in dendritic arbor formation. Primary hippocampal neurons, which co-expressed RNAi#1 with JMY*ΔCaM, showed numbers of dendritic branch points and dendritic terminal points as well as values for total dendritic length that were as low as for RNAi#1 expressing neurons, i.e., significantly lower than control (scr. RNAi) or those for rescue with JMY*, for all dendritic parameters (Fig. 5H–J) although also this mutant was provided in excess and was available in soma and dendrites (Fig. 5K). Sholl analyses also demonstrated that JMY*ΔCaM failed to rescue JMY deficiency phenotypes (Supplementary Fig. 5E).

These results demonstrated that the deletion of the region encompassing the CaM binding site in the JMY C-terminus strongly affected the functionality of JMY in neuromorphogenesis.

## The first WH2 domain of JMY strongly binds to actin and is essential for dendritic branching

Due to the partial overlap of the identified CaM binding site in the C-terminus of JMY with JMY's WH2 domains and the strict requirement of this region for the functions of JMY in neuromorphogenesis (Figs. 3, 4 and 5), we next aimed at studying the molecular and functional properties of JMY's WH2 domains more closely.

In vitro assays with purified recombinant individual WH2 domains of JMY showed that in principle, all three of the JMY WH2 domains were able to bind to G-actin directly. All three WH2 domains recruited fluorescently labeled actin at bead surfaces in a specific manner (Fig. 6A) and inhibited actin filament formation in in vitro-pyrene actin polymerization assays (Fig. 6B). Coprecipitation studies confirmed that each of JMY's three WH2 domains was functional and able to precipitate endogenous actin from rat brain lysate based on classical actin/WH2 domain interactions, as introduction of point mutations according to Zuchero et al.[26] (WH2[mut]) completely abolished binding to G-actin (Supplementary Fig. 6A). Interestingly, in comparison, JMY WH2#1 appeared to show the strongest actin binding (Fig. 6A, B and Supplementary Fig. 6A).

In order to address the in vivo-relevance of the actin interactions of the three different WH2 domains, we next established a specific coimmunoprecipitation of endogenous actin from L929 cells with JMY's entire C terminal actin binding and nucleating domain (GFP-JMY CT) (Fig. 6C). Strikingly, under the same conditions, only immunoprecipitations of JMY WH2#1 but not immunoprecipitations of either WH2#2 or WH2#3 resulted in efficient coimmunoprecipitation of endogenous actin, as also shown by comparative quantitative analyses. The similarity of the actin coimmunoprecipitation levels by JMY CT and by JMY WH2#1 suggested that the actin association seen in coimmunoprecipitations mostly reflected the contribution of the first WH2 domain (Fig. 6C, D).

These results suggested that CaM binding may indeed have the power to significantly modulate the loading of JMY with G-actin, as $Ca^{2+}$-activated CaM bound to the WH2 domain with the highest in vivo relevance of interaction. In order to test whether this also correlated with a high functional relevance in developing neurons, we transfected developing primary hippocampal neurons with JMY RNAi and a JMY* mutant solely deficient for actin binding via WH2 domain #1 (JMY*[WH2#1mut]) and examined the effects on dendritic arbor formation (Fig. 6E–H, Supplementary Fig. 6B, C). Interestingly, JMY*[WH2#1mut] was unable to rescue any of the JMY RNAi#1-mediated knockdown phenotypes. Neurons co-expressing RNAi#1/ JMY*[WH2#1mut] had significantly lower dendritic branch points, dendritic terminal points, total dendritic length, and dendritic Sholl intersection compared to scr. RNAi control and the quantitative values determined for all of these parameters were almost indistinguishable from neurons expressing RNAi#1, although also this mutant was also provided in excess (Fig. 6E–H and Supplementary Fig. 6B–D).

Thus, despite the availability of two further WH2 domains, interfering solely with the actin binding of JMY's first WH2 domain was sufficient to render JMY inactive in dendritic arbor formation.

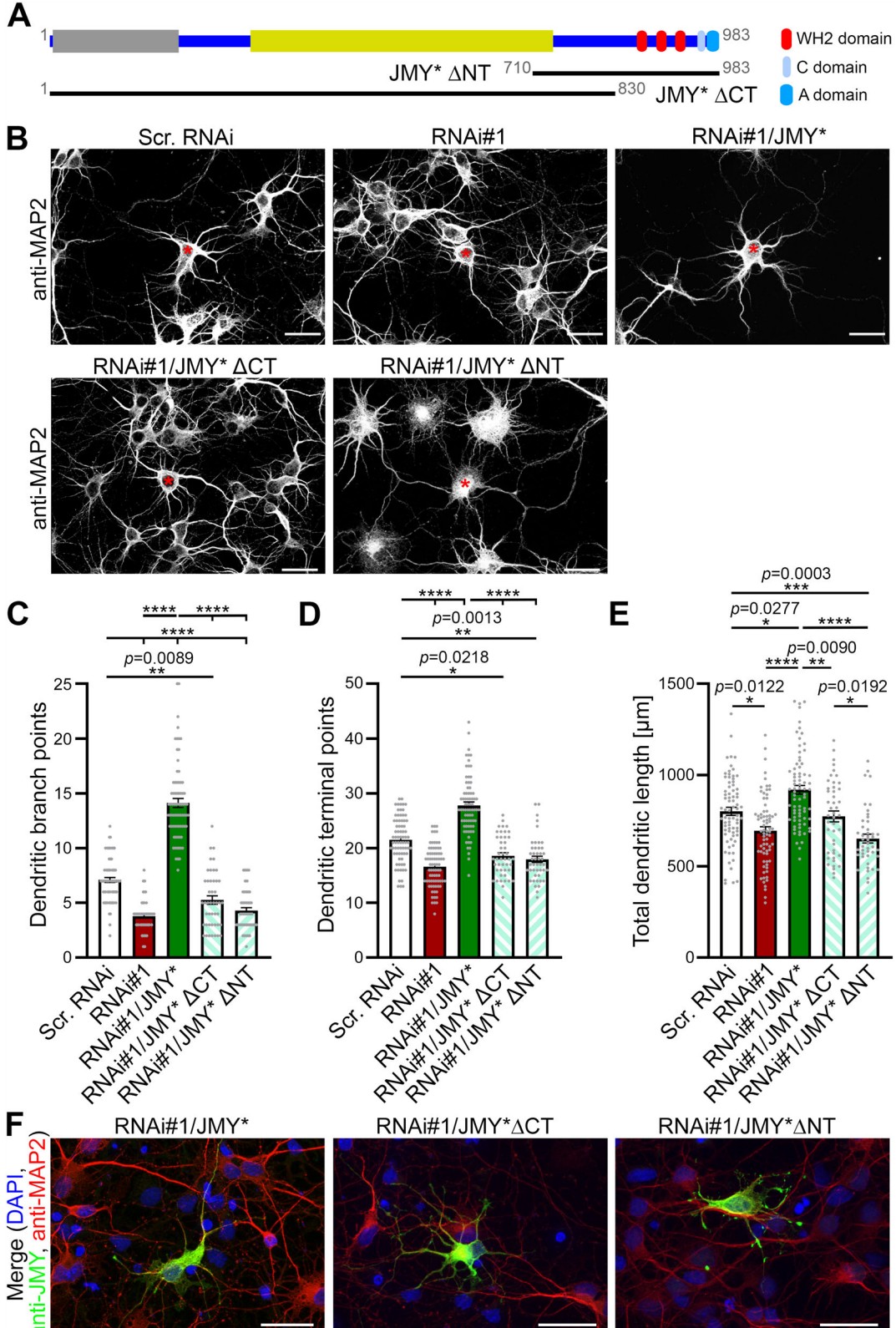

### The actin association of JMY's first WH2 domain is controlled by binding of Ca²⁺-activated CaM

To explore the possible relation between $Ca^{2+}$/CaM-signaling and the binding abilities of JMY to actin, we set up in vitro reconstitutions with purified proteins. These experiments showed that addition of $Ca^{2+}$ promoted the association of CaM with a part of JMY comprising its first WH2 domain and flanking regions (WH2#1) (Fig. 6I, J). Additionally, these analyses revealed that CaM association decreased the ability of the first WH2 domain of JMY to associate with actin (Fig. 6I, J). The observed modulation of the actin binding of the first WH2 domain of JMY required the $Ca^{2+}$ sensor protein CaM, as it did not occur upon addition of $Ca^{2+}$ alone (Supplementary Fig. 6E, F).

Both the increase of CaM binding as well as the decline of actin binding were statistically significant when conditions with and without $Ca^{2+}$ were

**Fig. 3 | Development of the dendritic arbor depends on both the N- and C-terminus of JMY. A** Schematic representation of JMY and mutant variants thereof used to rescue JMY loss-of-function phenotypes. **B** MIPs of anti-MAP2-immunostained primary hippocampal neurons transfected at DIV4 with either scr. RNAi, RNAi#1, RNAi#1/JMY*, RNAi#1/JMY* ΔCT, and RNAi#1/JMY* ΔNT, respectively, and fixed 30 h after transfection. Transfected cells are expressing GFP as reporter (see Supplementary Fig. 3A) and are marked by asterisks. Quantitative analyses of dendritic branch points (**C**), dendritic terminal points (**D**), and total dendritic length (**E**). In comparison to neurons transfected with RNAi#1, only coexpression of wild-type JMY* but not JMY* ΔCT or JMY* ΔNT rescued the JMY loss-of-function phenotypes shown by dendritic branch point, dendritic terminal point, and dendritic length analyses. **F** MIPs of hippocampal neurons transfected and fixed under the same conditions as for the morphometric analyses (**B–E**) with additional anti-JMY immunostainings and DAPI stainings showing that the mutants not able to rescue were successfully expressed and available in soma and dendrites. Bars, 30 μm. Data, mean ± SEM shown as bar/dot plots. Scr. RNAi, RNAi#1 and RNAi#1/JMY* $n = 75$ neurons (partially included in Fig. 2, too); RNAi#1/JMY* ΔNT and RNAi#1/JMY* ΔCT, $n = 45$ neurons each from 2 to 5 independent assays. Numerical data are provided in Supplementary Data 1. Significant differences, One-way ANOVA/Tukey's post-test; *$p < 0.05$, **$p < 0.01$, ***$p < 0.001$, ****$p < 0.0001$. For exact $p$ values see figure panels. Note that ****$p < 0.0001$ are too small values to be reported by the software used.

compared in quantitative, fluorescence-based Western blotting analyses (Fig. 6J).

In line with these in vitro reconstitution results, coimmunoprecipitation studies subjected to quantitative Western blotting also uncovered a $Ca^{2+}$-mediated control of the actin binding capabilities of JMY. Importantly, not only the individual JMY WH2#1 but also the entire C-terminal JMY part comprising all three WH2 domains and the Arp2/3 complex interface showed a strongly impaired coimmunoprecipitation of endogenous actin in the presence of 2 μM $Ca^{2+}$ compared to $Ca^{2+}$-free conditions (Fig. 6K). For both examined proteins, quantitative analyses of coimmunoprecipitated actin signals demonstrated a highly significant and very severe reduction in JMY's actin coimmunoprecipitation ability under 2 μM $Ca^{2+}$ compared to their respective $Ca^{2+}$ free condition (more than −80%) (Fig. 6L).

Incubation with 2 μM $Ca^{2+}$ for 1 h and subsequent addition of EGTA to mimic the decline of a temporary $Ca^{2+}$ signal demonstrated that the suppression of GFP-JMY CT's association with endogenous actin in the presence of 2 μM $Ca^{2+}$ was partially reversible by the EGTA addition. Quantitative analysis of the coimmunoprecipitated actin signals demonstrated a significant $Ca^{2+}$-mediated decrease of the actin binding ability of GFP-JMY CT (about −80%) and a partial but significant restauration of actin binding upon subsequent quenching of $Ca^{2+}$ using EGTA (Fig. 6M and Supplementary Fig. 6G).

Taken together, $Ca^{2+}$ signals and CaM as $Ca^{2+}$ sensor protein regulate the interaction between JMY and actin, and this regulation of JMY's actin binding is mediated by the first WH2 domain of JMY.

### $Ca^{2+}$/CaM singalling is critical for JMY-mediated neuromorphogenesis

Finally, it was important to examine whether the functions of JMY in dendritic arborization would not only require the CaM binding site of JMY but also functionally explicitly rely on $Ca^{2+}$/CaM-signaling. We therefore addressed whether the CaM inhibitor CGS9343B[27] would affect JMY-mediated neuromorphological functions in primary neurons (Fig. 7A–E). CGS9343B was reported to lead to some reductions in dendritic branch points, in the frequency of protrusion initiation, and in the dynamics of neuronal structures, resulting in a more static morphology during dendritic arbor development[8]. Developing primary hippocampal control neurons treated with CGS9343B indeed showed reduced numbers of dendritic branch points as well as minor reductions of terminal point numbers (Fig. 7B, C). This observed inhibition is in line with previous observations[8,17], and most likely reflects a variety of mostly still unknown effectors being addressed by CaM.

Strikingly, while the subcellular localization of JMY* did not change in any obvious manner (Supplementary Fig. 7), the CaM inhibitor CGS9343B suppressed the JMY*-mediated effects on dendritic branching. Neurons expressing JMY*/GFP and treated with CGS9343B showed significantly reduced dendritic branch point numbers, dendritic terminal point numbers, total dendritic length, and dendritic Sholl intersections compared to JMY*/GFP neurons without CGS9343B treatment (Fig. 7A–E). The observed suppression of JMY-mediated dendritic arborization was so strong that dendritic branch points, terminal points, and total dendritic length of CaM inhibitor CGS9343B-treated control neurons expressing solely GFP and of neurons expressing JMY*/GFP were at similar levels (Fig. 7B–D). The suppression of all JMY-mediated dendritic arbor formation processes by the CaM inhibitor CGS9343B was thus absolute. This emphasized the importance of $Ca^{2+}$/CaM signaling for JMY-mediated dendritic arbor development (Fig. 8).

## Discussion

A prerequisite for proper neuronal network formation in the brain is the ability of neurons to develop into polar and highly arborized morphologies. This process is thought to rely on signals converted into locally restricted and transient activities of force-generating cytoskeletal effectors[4–6,18,28]. The formation of new dendritic branches hereby coincides with local $Ca^{2+}$ signals and F-actin formation at dendritic branch induction sites[8]. The functions of the WH2 domain-based actin nucleator Cobl[9] in dendritogenesis are tightly regulated by the $Ca^{2+}$ sensor protein calmodulin (CaM)[8,10]. Our identification of the WH2 domain-based actin nucleator JMY[13,15] as a direct binding partner and target of CaM now uncovers that $Ca^{2+}$/CaM-mediated control of actin filament formation does not represent a special feature of Cobl but a more generally used, critical principle among WH2 domain-based actin nucleators (Fig. 8).

The relevance of the JMY identification as CaM target is strongly supported by coprecipitations of endogenous JMY, coimmunoprecipitation of the two endogenous proteins, and corecruitment studies in intact cells. The $Ca^{2+}$ dependency of the JMY interaction clearly suggests that JMY/CaM complexes translate $Ca^{2+}$ signals sensed by CaM into cellular answers. Spatial and temporal fine-control by $Ca^{2+}$/CaM signaling is thought to be of utmost importance in the brain, but the effectors executing the different cell biological functions are only beginning to emerge.

JMY is unique among the members of the rather new family of WH2 domain-based actin nucleators, as JMY can nucleate actin both directly via its three G-actin-binding WH2 domains and indirectly via the Arp2/3 complex[13,15]. JMY was shown to be important for blastocyst development of mice and pigs[29,30] but also is highly expressed in the brain[15]. We show that JMY is expressed in neurons and identify a critical role of JMY in the dendritic arbor formation of neurons.

Surprisingly, despite its ability to promote actin filament formation, JMY was introduced as negative regulator of neurite outgrowth in Neuro 2a cells[15]. In case observations in this cell line would reflect mechanisms of neuronal development, this raised the question whether $Ca^{2+}$ signals decoded by CaM—via interfacing with different actin nucleators—either regulate distinct aspects or maybe even work in opposite directions in a given aspect in neuronal development. Our extensive gain- and loss-of-function studies for JMY in developing primary hippocampal neurons clearly showed that this is not the case. Our data reveal that JMY is a positive regulator of dendritogenesis. An excess of JMY resulted in a strong increase of dendritic branching. Conversely, JMY RNAi led to a strong impairment in dendritic arborization. Dendritic branch depth analyses revealed that the number of primary dendrites was rather unaffected by changes in JMY

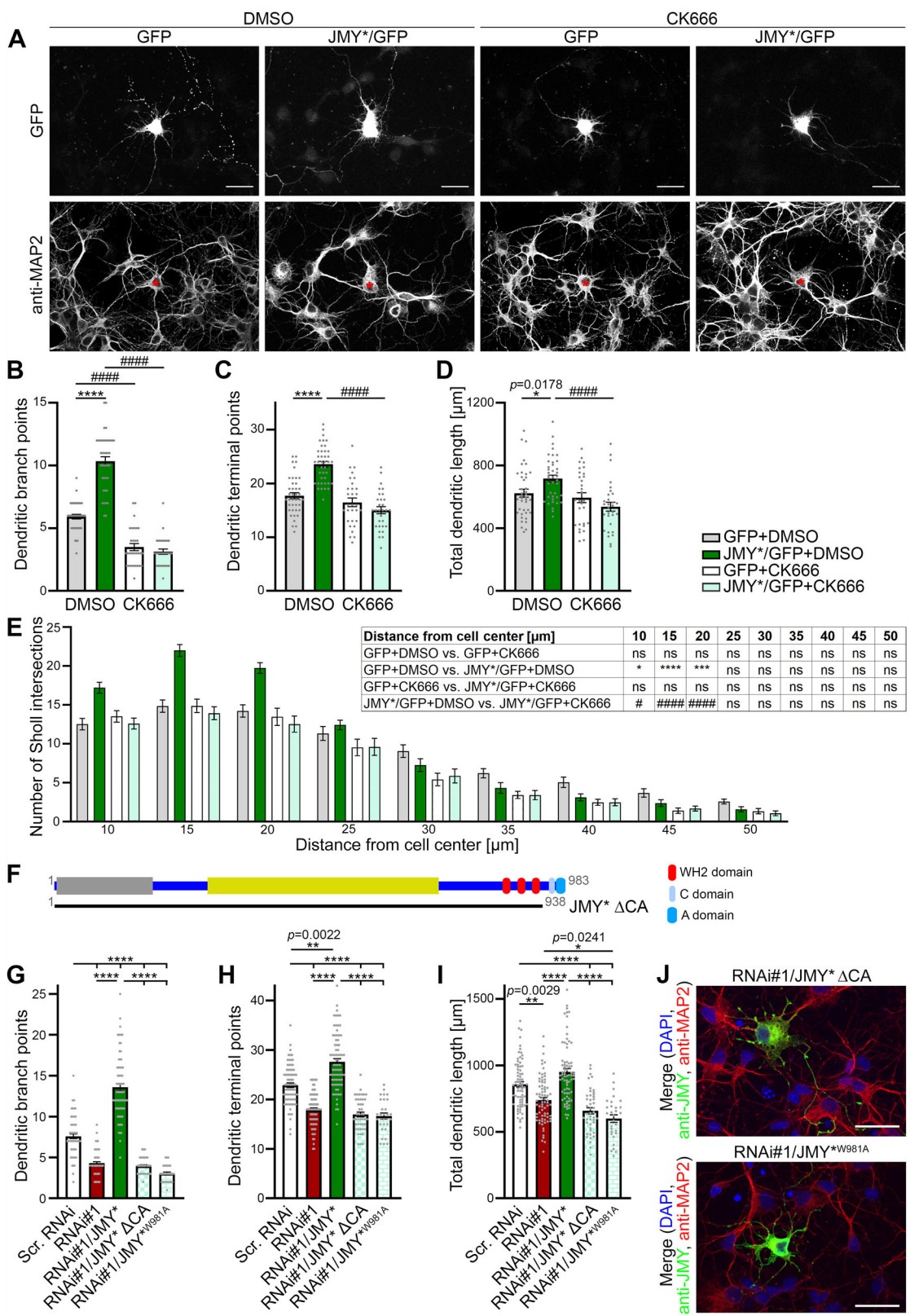

expression levels. Instead, JMY was required for branch formation from existing primary dendrites and thus plays a critical role in dendritic arborization.

The successful rescue of all identified JMY deficiency phenotypes in dendritic arbor formation by expression of untagged RNAi-insensitive JMY* clearly demonstrated that the RNAi-mediated effects were JMY-specific. Promotion of dendritic arborization thus represents the physiological function of JMY in neurons.

Some reports have suggested an inhibitory, potentially autoinhibitory mechanism for N-terminally tagged versions of full-length JMY in cells[15,31]. In contrast, our gain-of-function and particularly our rescue experiments in loss-of-function studies demonstrate that, in neurons but also in COS-7

**Fig. 4 | Arp2/3 complex activation and binding by JMY is essential for dendritic arbor formation of developing neurons. A** MIPs of anti-MAP2 immunostained and GFP-expressing (reporter) primary hippocampal neurons transfected with plasmids indicated (transfected cells marked by asterisks) at DIV4 and treated with the Arp2/3 complex inhibitor CK666 and the solvent DMSO, respectively, 24 h later and fixed 30 h after transfection. Bars, 30 μm. Quantitative analyses of dendritic morphologies revealing that the JMY*/GFP + DMSO-mediated increases in dendritic branch points (**B**), dendritic terminal points (**C**), and total dendritic length (**D**) compared to GFP+DMSO control neurons were completely abolished upon addition of the Arp2/3 complex inhibitor CK666. **E** Sholl analyses show that the complete loss of JMY-mediated dendritic arbor complexity increases upon CK666 application occurs throughout the dendritic arbor. **F** Schematic representation of JMY and of a mutant lacking the Arp2/3 complex-binding CA domain. **G–I** Quantitative analyses of dendritic morphology of neurons transfected with RNAi (control), RNAi#1 against JMY or RNAi#1/JMY*, RNAi#1/JMY* ΔCA or RNAi#1/JMY*[W981A] showing that, in contrast to the successful rescue with JMY*, reexpression of JMY* ΔCA or JMY*[W981A] being unable to bind and/or activate the Arp2/3 complex were incapable of rescuing the RNAi#1-induced decrease in dendritic branch points (**G**),

dendritic terminal points (**H**) and dendritic length (**I**) in comparison to scr. RNAi. (**J**) MIPs of hippocampal neurons transfected and fixed under the same conditions as for the morphometric analyses (**G–I**) with additional anti-JMY immunostainings and DAPI stainings showing that the mutants not able to rescue were successfully expressed and available in soma and dendrites. Bars, 30 μm. Data, mean ± SEM shown as bar/dot plots (**B–D, G–I**) and bar plot (**E**), respectively. **B–E** GFP+DMSO, $n = 40$; JMY*/GFP+DMSO, $n = 40$; GFP+CK666, $n = 30$; JMY*/GFP+CK666, $n = 30$ cells from two independent assays. **G–I** Scr. RNAi., RNAi#1 and RNAi#1/ JMY* (part of the data were already included in Fig. 2B-D), $n = 75$; RNAi#1/JMY* ΔCA, $n = 45$; RNAi#1/JMY*[W981A], $n = 30$ neurons from 2 to 5 independent assays. Numerical data are provided in Supplementary Data 1. Statistical assessments were conducted by comparing the factor transfection (GFP vs. JMY*/GFP, *) or treatment (DMSO vs. CK666, #) using Two-way ANOVA, Sidak's test (**B–D**) as well as Three-way ANOVA with Tukey's multiple comparisons test (**E**) and by using One-way ANOVA/Tukey's post-test (**G–I**), respectively. * and #$p < 0.05$, **$p < 0.01$, ***$p < 0.001$, **** and ####$p < 0.0001$. For exact p values see figure panels. Note that ****$p < 0.0001$ are too small values to be reported by the software used.

---

cells, untagged full-length versions of JMY showed full activity in cellular morphogenesis. Somewhat in line with our identification of a morphogenesis-promoting role of JMY in neurons, JMY induced F-actin-rich 3D-membrane ruffles in COS-7 (this study), and JMY knockdown prevented cellular differentiation in oligodendrocyte progenitor cells[16]. In osteosarcoma-derived cell lines and in HUVEC but not in NIH3T3 cells, a role of JNY in cell motility was observed[13–15,32]. Together, these discrepancies may reflect cell-type-specific effects in the functions of JMY.

Our work clearly demonstrated a critical role of JMY in neuronal development and, in addition, highlighted which molecular mechanisms are critical in bringing about the identified functions of JMY in dendritic arbor formation (Fig. 8). JMY can produce unbranched actin filaments directly through its G-actin binding WH2 domains and can furthermore promote the nucleation of branched actin filaments via Arp2/3 complex activation in in vitro actin filament formation assays[13,15,19]. Our functional examinations revealed that JMY's functions in neuronal development clearly required both the N-terminal domains, which by some yet to be clarified mechanism seems to support JMY's actin nucleation activity[20] but were also reported to associate with p300 and STRAP and thereby link JMY to nuclear DNA damage responses[11,33], and depended on C terminal WH2 domain-mediated actin binding and Arp2/3 complex coupling.

Application of the Arp2/3 complex inhibitor CK666[22,23] as well as our functional analyses with two different JMY mutants incapable of associating with the Arp2/3 complex clearly demonstrate the importance of Arp2/3 complex association and activation in JMY-mediated functions in dendritic arbor formation.

In line, knockdown of Arp2/3 complex components led to reductions in dendritic branch number, density, and length in *Drosophila* sensory class IV neurons[34] and also led to a complete, albeit mechanistically not yet fully understood suppression of Cobl-mediated dendritic arborization in rat hippocampal neurons[35].

Importantly, our functional analyses revealed that also JMY's WH2 domain-mediated G-actin binding abilities that underlie JMY's intrinsic actin nucleation activity[13] were essential for dendritic arbor formation. Importantly, even impairing solely the actin binding of JMY's first WH2 domain was sufficient to render JMY completely inactive in dendritic arborization, even though two further WH2 domains were still present. In vitro-actin polymerization assays demonstrated that mutating the first and second WH2 domains of JMY together completely prevented actin nucleation activity of JMY, but still allowed for Arp2/3 complex activation[26]. In contrast, efficient actin nucleation by JMY's intrinsic nucleation activity required all three WH2 domains[26]. Our functional data thus suggest that JMY powers dendritic arbor formation by a combination of its different actin nucleation mechanisms (Fig. 8).

Strikingly, the first WH2 domain of JMY coincided with a CaM-binding region of JMY, and several lines of evidence clearly showed the

impact of CaM association and $Ca^{2+}$/CaM signaling on JMY. Deletion of the CaM binding site in the JMY C-terminus strongly impaired JMY functions in neurons. Interestingly, coimmunoprecipitation of endogenous actin from cellular lysates demonstrated that the first WH2 domain of JMY represents the strongest and most efficient G-actin-binding module. The identified CaM binding site overlapped with this WH2 domain, and $Ca^{2+}$-dependent CaM association completely suppressed the actin association of the first WH2 domain. In vitro reconstitutions demonstrated that this was a direct effect of $Ca^{2+}$-activated CaM on JMY's first WH2 domain and that $Ca^{2+}$ alone had no effect. Importantly, quantitative coimmunoprecipitations of endogenous actin revealed that this $Ca^{2+}$/CaM control of JMY's actin loading not only had profound impact on the first WH2 domain but was similarly observed for the entire C-terminal part of JMY. The actin association seen in JMY coimmunoprecipitation studies thus mostly seems to reflect the contribution of the first, $Ca^{2+}$/CaM-regulated WH2 domain, implying profound impact of $Ca^{2+}$/CaM signaling on JMY's cytoskeletal functions.

Interestingly, the suppression of the actin binding of JMY was partially reversed upon falling $Ca^{2+}$ concentrations. Thus, JMY-mediated promotion of actin filament formation may be attenuated at the peak of a $Ca^{2+}$ transient but is not hampered at lower $Ca^{2+}$ levels. It is possible that such a transient suspension of local actin filament formation may be related to the proposed mechanism of calcium-mediated actin reset (CaAR) mediating acute cellular adaptations in response to signals and stress proposed in non-neuronal cells[36]. Our observations that the CaM inhibitor CGS9343B completely suppressed all aspects of JMY-promoted dendritic arbor formation strongly suggest that $Ca^{2+}$/CaM-signaling-mediated fine-tuning of its actin loading is a key aspect of JMY's functions.

Suppression of actin loading by $Ca^{2+}$-activated CaM, inhibiting the action of a distinct WH2 domain was also observed for the actin nucleator Cobl[8]. Therefore, the actin nucleators JMY and Cobl may share a WH2 domain-based actin nucleation mechanism[37] that is controlled by a similar mode of $Ca^{2+}$/CaM-dependent fine-tuning of a single WH2 domain in an actin nucleation-competent set of three WH2 domains (Fig. 8). In contrast, in Cobl's anchester Cobl-like, which is different from Cobl as it does not seem to nucleate actin by itself but works in a complex with the F-actin-binding protein Abp1, CaM control does not converge on the single WH2 domain of Cobl-like but on the Abp1 interaction site[16].

Interestingly, $Ca^{2+}$/CaM-signaling convering onto the actin nucleators Cobl and JMY also includes distinct, peculiar aspects. Apart from the transient suppressive effect of $Ca^{2+}$/CaM on one of their WH2 domains, we identified for both JMY and Cobl, Cobl showed an overall increased G-actin binding upon elevated $Ca^{2+}$ levels. $Ca^{2+}$/CaM furthermore controlled Cobl's targeting to the plasma membrane[8]. Additionally, pathophysiological $Ca^{2+}$ levels caused a proteolytic, calpain-mediated breakdown of Cobl during the hours subsequent to an ischemic stroke[38].

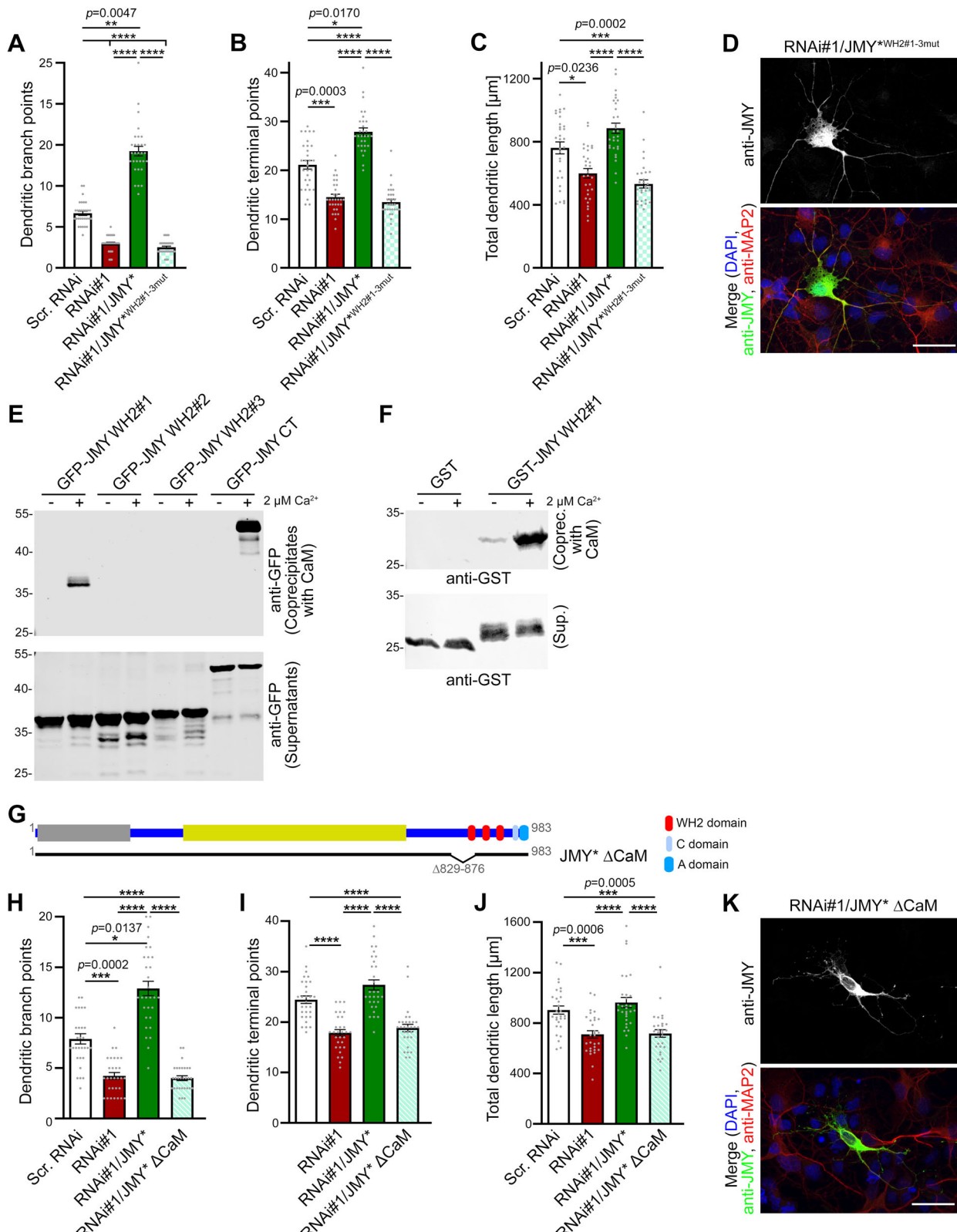

Our identification of JMY functions in neurons and of their control by $Ca^{2+}$/CaM furthermore suggests that $Ca^{2+}$/CaM-controlled actin filament formation is a thus far underrecognized, general, and critical but multifaceted principle in cell biology that is of much more widespread importance for cytoskeletal control and for shaping cells—particularly neurons—than initially thought.

# Methods

## DNA constructs

Plasmids encoding for GFP- and untagged JMY (Mus musculus JMY, NP_067285.2; GI: 61098108) and parts thereof were generated by PCR by using a pLAP-JMY construct received from the Welch lab (University of California, Berkeley) as template (for a complete list of primers, please see

**Fig. 5 | JMY's functions in dendritogenesis rely on actin-binding and on a CaM binding region comprising the first WH2 domain of JMY.** Quantitative analyses of dendritic branch points (**A**), dendritic terminal points (**B**), and total dendritic length (**C**) of anti-MAP2-immunostained primary hippocampal neurons transfected at DIV4 with either scr. RNAi, JMY RNAi#1, or with JMY RNAi#1 together with RNAi insensitive JMY* and JMY*[WH2#1-3mut], respectively, and fixed 30 h after transfection. Note that JMY*[WH2#1-3mut] failed to rescue any of the JMY loss-of-function phenotypes. **D** MIP of hippocampal neurons transfected and fixed under the same conditions as for the morphometric analyses (**A–C**) with additional anti-JMY immunostaining and DAPI staining showing that JMY*[WH2#1-3mut], albeit not able to rescue, was successfully expressed and available in soma and dendrites. **E** Immunoblot of GFP-JMY deletion mutants expressed in L929 cells coprecipitated with a CaM matrix. GFP-JMY CT and GFP-JMY WH2#1 were coprecipitated with CaM matrix under 2 μM Ca²⁺ (+), but not under Ca²⁺ free (−) conditions. The anti-GFP signal of the supernatants confirmed the expression and integrity of GFP and GFP-fusion proteins in the assay. **F** Coprecipitation of purified GST-JMY WH2#1 but not GST with immobilized CaM demonstrating that the Ca²⁺-dependent interaction of JMY with CaM is direct. **G** Schematic representation of JMY and of a mutant lacking the CaM binding region. Quantitative analyses of dendritic branch points (**H**), dendritic terminal points (**I**), and total dendritic length (**J**) of anti-MAP2-immunostained primary hippocampal neurons transfected at DIV4 and fixed 30 h thereafter. Note that JMY*ΔCaM failed to rescue the impairments in dendritic arborization caused by JMY RNAi. **K** MIP of hippocampal neurons transfected and fixed under the same conditions as for the morphometric analyses (**H–J**) with additional anti-JMY immunostaining and DAPI staining showing that JMY*ΔCaM, albeit not able to rescue, was successfully expressed and available in soma and dendrites. Bars, 30 μm. Data, mean ± SEM shown as bar/dot plots. **A–C** Scr. RNAi, RNAi#1, RNAi#1/JMY* and JMY*[WH2#1-3mut] $n = 30$ neurons from 2 independent assays. **H–J** Scr. RNAi, RNAi#1, RNAi#1/JMY*, and RNAi#1/JMY*[ΔCaM] $n = 30$ neurons from two independent assays (data for the first three conditions also included in Fig. 4G–I). Numerical data and images of uncropped and unedited blots are provided in Supplementary Data 1 and 2, respectively. Statistical significances, One-way ANOVA/Tukey's post-test (**A–C, H–J**). *$p < 0.05$, **$p < 0.01$, ***$p < 0.001$, ****$p < 0.0001$. For exact $p$ values see figure panels. Note that ****$p < 0.0001$ are too small values to be reported by the software used.

Supplementary Table 1). The internal HindIII site was used for subcloning and fusing N-terminal and C-terminal (mutant) parts. Usually, JMY and its deletion and/or point mutants were cloned first into pEGFP (Clontech).

Constructed were GFP-JMY full-length (aa1-983), GFP-JMY ΔCT (aa1-830) and GFP-JMY CT (aa821-983), as well as GFP-JMY WH2#1 (aa841-883), GFP-JMY WH2#2 (aa874-914) and GFP-JMY WH2#3 (aa904-947), GFP-JMY ΔNT (aa710-983) and GFP-JMY ΔNT ΔCaM (aa710-828 + 877-983) in pEGFP-C vectors (Clontech). GFP-JMY ΔNT ΔCaM was generated exploiting an internal PstI site (corresponding to amino acid position 828).

The individual JMY WH2 domains were also subcloned into pGEX-4T1 (GE Healthcare) (GST-JMY WH2#1; GST-JMY WH2#2; GST-JMY WH2#3). Corresponding mutants with disrupted G-actin binding were generated by PCR with primers inducing the mutations L857A, F858A, and L870A (GST-JMY WH2#1[mut]), V889A, L890A, and L900A (GST-JMY WH2#2[mut]), and I920A, L921A, and L930A (GST-JMY WH2#3[mut]), respectively. For a list of the mutation primers, please see Supplementary Table 1.

All functional work was conducted with untagged JMY, which additionally carried several silent mutations rendering JMY insensitive to attack by RNAi#1 and RNAi#2 (JMY*; nt 69-96 silently mutated). For the mutation primers used to generate JMY*, please see Supplementary Table 1. Plasmids expressing untagged JMY* were generated by inserting JMY* into pIRES2-EGFP (Clontech). Using an IRES to drive GFP expression, the resulting plasmid JMY*/GFP thus expresses untagged JMY* and GFP as an independent reporter from the same mRNA. pIRES2-EGFP was used as control for JMY*/GFP overexpression experiments in both primary hippocampal neurons and COS-7 cells.

RNAi tools directed against mouse and rat JMY were built using primer annealing[10] and insertion into pRNAT-H1.1 (GenScript). RNAi#1 corresponded to nt 69–89 (for primers see Supplementary Table 1). RNAi#2 corresponded to nt 78–98 (for primers see Supplementary Table 1). A scrambled RNAi sequence-containing pRNAT-H1.1 (scr. RNAi)[39] served as control.

Since rescue experiments required the additional expression of untagged JMY* and GFP as fluorescent reporter, we incorporated a new multiple cloning site into the intermediate scr. RNAi, RNAi#1, and RNAi#2 constructs using primer annealing (for primers see Supplementary Table 1) and then additionally inserted IRES-GFP from pIRES2-EGFP (Clontech) to generate suitable RNAi tools for functional experiments.

The generated IRES-GFP-reported scr. RNAi, RNAi#1, and RNAi#2 plasmids then allowed for insertion of JMY* and mutants thereof by subcloning into the newly inserted multiple cloning site to obtain RNAi rescue constructs. RNAi#1/JMY* ΔCT (aa1-830), RNAi#1/JMY* ΔNT (aa710-983), and RNAi#1/JMY* ΔCaM (aa1-828 + 877-983) were generated by subcloning the respective mutants (generated first as pEGFP constructs) into pRNAT-H1.1-IRES-GFP containing RNAi#1 and IRES-GFP. RNAi#1/JMY* ΔCA (aa1-938) and RNAi#1/JMY*[W981A] were generated by PCRs and insertion into RNAi#1. Versions of JMY* containing mutated WH2 domains were constructed by first mutating the respective WH2 domain(s) in a context encoding for the C-terminal part of JMY and then fusing these mutated pieces with an N-terminal part-encoding fragment of JMY* using an internal HindIII site. We generated an RNAi rescue construct with all three WH2 domains mutated (RNAi#1/JMY*[WH2#1-3mut] (aa1-983 with L857A, F858A, L870A, V889A, L890A, L900A, I920A, L921A, L930A)) and an RNAi rescue construct with only the first WH2 domain mutated (RNAi#1/JMY*[WH2#1mut] (aa1-983, L857A, F858A, L870A)). See Supplementary Table 1 for primers used for the different PCRs.

All segments of constructs generated by PCR or by primer annealing experiments were verified by sequencing.

GST-Cobl WH2#2 (aa1206-1277) was as described in ref. [9].

Mito-mCherry-CaM and Mito-mCherry also were as described in ref.[8].

In order to be able to express and purify untagged, fluorescently labeled CaM protein, CaM was subcloned from Mito-mCherry-CaM into pGEX-6P (GE Healthcare) —a vector that allows for tag removal from recombinant proteins using PreScission Protease.

### Antibodies, reagents, and proteins

Rabbit anti-JMY antibodies directed against the C-terminal part of JMY, applied for detecting the expression and localization of untagged JMY* ΔNT, were from Proteintech Group, Inc. (25098-1-AP). Rabbit anti-JMY antibodies directed against the N-terminal part of JMY were from Antibodies-online GmbH (ABIN2838360). Polyclonal rabbit anti-GFP (ab290) and polyclonal chicken anti-GFP (ab13970) antibodies, as well as monoclonal mouse anti-Arp3 antibodies (ab49671), were from Abcam. Monoclonal mouse anti-GFP antibodies (JL-8) were from Clontech (632381). Monoclonal anti-β-actin (A5441), anti-α-actin (A2066), anti-MAP2 (HM-2; M4403) and anti-β-tubulin antibodies (T4026) were from Sigma. Monoclonal mouse anti-CaM (G3) antibodies were from Santa Cruz Biotechnology. Affinity-purified rabbit and guinea pig anti-GST antibodies were described previously[40,41].

Alexa Fluor 568-conjugated phalloidin and MitoTracker Deep Red 633 were from Molecular Probes.

Secondary antibodies used included Alexa Fluor 568-labeled donkey anti-mouse antibodies (R37114 and A10037), Alexa Fluor 680-labeled goat anti-mouse (A-21236 and A32723) and anti-rabbit antibodies (AB_2535758), Alexa Fluor-labeled 488 donkey anti–rabbit (A-21206), Alexa Fluor 680-labeled donkey anti-rabbit (AB_2922888), goat anti-rabbit antibodies coupled to IRDye800 (AB_2556775) and DyLight800-conjugated goat anti–mouse antibodies (AB_2556774) were from Thermo Fisher Scientific. Donkey anti-guinea pig antibodies coupled to

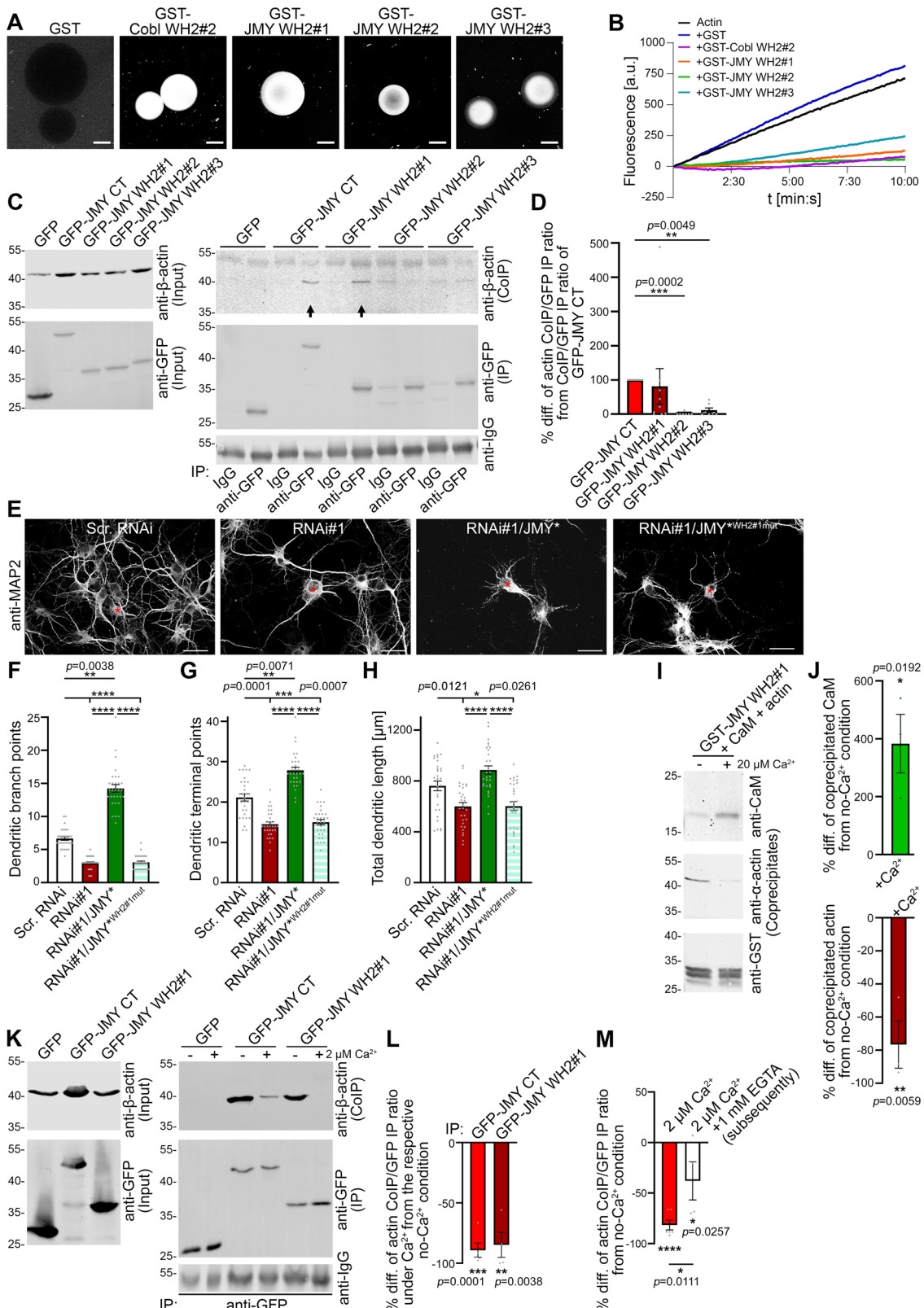

IRDye680 (925-32411) and donkey Anti-chicken IRDye 800CW (925-32218) were from LI-COR Bioscience. Goat anti-chicken Alexa Fluor 680 was from Abcam.

Sepharose 4B-coupled CaM was from GE Healthcare. Rabbit skeletal muscle actin and pyrene-labeled rabbit skeletal muscle actin were from Cytoskeleton. Alexa-Fluor 568-labeled rabbit skeletal muscle actin was

generated by using the Alexa-Fluor 568 protein labeling kit from Thermo Fisher Scientific.

**Purification of recombinant proteins and tag cleavage**
GST-tagged fusion proteins were purified from *E. coli*[35,42]. In brief, GST-tagged proteins were expressed in *E. coli* BL21 and purified from bacterial

**Fig. 6 | JMY's first WH2 domain interacts with actin most strongly, is key in JMY-mediated dendritic arbor formation, and is controlled in a CaM/Ca²⁺-dependent manner. A** Fluorescence microscopy image of specific recruitment of purified, Alexa Fluor™ 568-labeled actin by purified, individual GST-JMY WH2 domains immobilized to beads (visualized live 30 min after incubation start). GST-Cobl WH2#2 (strong G-actin-binding WH2 domain) was used as positive control. Bars, 50 µm. **B** Measurements of fluorescence intensities during the polymerization of monomeric pyrene actin in absence and presence of individual WH2 domains of JMY. Cobl WH2#2 was used for comparison. Representative fluorescence intensity curves out of 3 assays conducted. Note that all three WH2 domains of JMY inhibited spontaneous actin polymerization, and that JMY WH2#3 had the lowest G-actin quenching effect when all three WH2 domains of JMY were compared to each other. **C, D** Immunoblot analyses of coimmunoprecipitation analyses of lysates of L929 cells expressing GFP or GFP-fusion proteins of JMY CT, JMY WH2#1, JMY WH2#2, and JMY WH2#3 (**C**; arrows mark specific coimmunoprecipitations of endogenous actin) and quantitative analyses thereof (**D**). The quantitative data are expressed as coimmunoprecipitated actin per immunoprecipitated GFP fusion protein normalized to the ratio obtained with GFP-JMY CT in the respective assay (in percent). Note that only GFP-JMY CT and GFP-JMY WH2#1 specifically coimmunoprecipitated endogenous actin. **E** Representative MIPs of anti-MAP2-stained primary hippocampal neurons transfected at DIV4 as indicated and fixed 30 h later. Transfected cells are marked by asterisks. Bars, 30 µm. **F–H** Quantitative analyses of dendritic morphology of primary hippocampal neurons transfected as indicated based on anti-MAP2 staining. The neurons showed JMY RNAi-mediated decreases in dendritic branch points (**F**), dendritic terminal points (**G**), and total dendritic length (**H**), which were rescued by reexpression of JMY* but not by JMY*^(WH2#1mut). **I** In vitro reconstitution of JMY WH2#1/CaM/actin complexes with purified proteins (GST-JMY WH2#1 immobilized) demonstrating the reciprocal Ca²⁺

dependence of the actin and of the CaM binding of JMY WH2#1. **J** Quantitative analyses of the JMY WH2#1/CaM/actin complex formation showing a highly significant decrease of actin binding and an increase of CaM binding in the presence of Ca²⁺. **K, L** Immunoblotting of coimmunoprecipitates of endogenous actin with GFP-JMY fragments expressed in L929 cells (**K**) and quantitative analyses thereof (**L**). The actin CoIP/GFP IP ratios expressed in percent difference from the respective Ca²⁺ free condition demonstrate a strong reduction of the actin binding of GFP-JMY CT as well as of GFP-JMY WH2#1 upon addition of 2 µM Ca²⁺ (+) compared to the Ca²⁺ free (EGTA containing) condition (−). **M** Quantitative analyses of coimmunoprecipitation experiments addressing the reversibility of the modulation of JMY's actin binding by 2 µM Ca²⁺ by subsequent chelation of the Ca²⁺ with EGTA (GFP-JMY CT was used for analyses). Data, mean ± SEM shown as bar/dot plots. **D** GFP-JMY CT, $n = 9$; GFP-JMY WH2#1, $n = 9$; GFP-JMY, WH2#2 $n = 7$; GFP-JMY WH2#3, $n = 7$ independent sets of coimmunoprecipitations. **F–H** Scr. RNAi, RNAi#1, RNAi#1/JMY* and JMY*^(WH2#1mut) $n = 30$ neurons from two independent assays (data partially repeated from Fig. 5). **J** $n = 3$ sets of comparisons. **L** GFP-JMY CT, $n = 5$ independent coimmunoprecipitation assays each; GFP-JMY WH2#1 $n = 4$ independent coimmunoprecipitation assays each. **M** GFP-JMY CT (EGTA), $n = 4$; GFP-JMY CT (Ca²⁺), $n = 6$; GFP-JMY CT (Ca²⁺ and subsequently EGTA), $n = 4$ independent quantitative coimmunoprecipitation assays. Numerical data and images of uncropped and unedited blots are provided in Supplementary Data 1 and 2, respectively. Statistical significances, Kruskal–Wallis test/Dunn's multiple comparison test (**D**), One-way ANOVA/Tukey's post-test (**F–H, M**), unpaired student's t-test (**J**), and two separate Welch's t-tests (**K**), respectively. *$p < 0.05$, **$p < 0.01$, ***$p < 0.001$, ****$p < 0.0001$. For exact p values see figure panels. Note that ****$p < 0.0001$ are too small values to be reported by the software used.

lysates using glutathione sepharose (Antibodies-Online GmbH) and eluted with 20 mM glutathione in 120 mM NaCl and 50 mM Tris, pH 8.0. After purification, fusion proteins were dialyzed against PBS.

Untagged recombinant CaM was obtained by adapting procedures established previously described in ref. 10. In brief, PreScission Protease (GE Healthcare) cleavage of GST-CaM obtained from expression of GST-CaM (driven by pGEX-6P) in *E. coli* BL21 was performed during dialysis against HN-buffer (150 mM NaCl, 2 mM DTT, and 20 mM HEPES (pH 7.4)) overnight at 4 °C. Subsequently, GST and remaining GST-CaM were removed from the reaction mixture by precipitation with glutathione-agarose (Antibodies-Online GmbH). Successful cleavage and protein integrity were verified by sodium dodecyl sulfate polyacrylamide gel electrophoresis (SDS-PAGE) and Coomassie staining.

Protein concentrations of purified proteins were determined by Bradford assays.

### In vitro reconstitution of direct protein/protein interactions
Direct protein/protein interactions were demonstrated by coprecipitation assays with combinations of immobilized GST-tagged JMY fusion proteins purified from *E. coli* and recombinant untagged CaM, as well as commercially available purified rabbit skeletal muscle α-actin, respectively. The coprecipitations were conducted in 10 mM HEPES pH 7.4, 250 mM NaCl, 0.1 mM MgCl₂, 1% (v/v) Triton X-100 supplemented with EDTA-free protease inhibitor cocktail as well as with 1 mM EGTA (no-Ca²⁺ condition) or with 20 µM Ca²⁺ (+Ca²⁺ condition), respectively. Incubations of GST-JMY proteins with actin, without and with Ca²⁺ were done to exclude putative direct Ca²⁺ effects on the actin binding of GST-JMY WH2#1.

Direct protein/protein interactions of GST-JMY WH2#1/GST with CaM were shown as described (in IP buffer containing 150 mM NaCl as well as with 1 mM EGTA (no-Ca²⁺ condition) or with 2 µM Ca²⁺ (+Ca²⁺ condition), respectively).

Eluted proteins were analyzed by SDS-PAGE and then subjected to immunoblotting with anti-α-actin, anti-CaM, and anti-GST antibodies. Primary antibodies were detected with fluorescent secondary antibodies using a Licor Odyssey System.

### In vitro actin recruitment assay
Actin recruitment assays were conducted with Alexa Fluor 568–labeled actin, an energy-regenerating mix, and immobilized GST-tagged fusion proteins[9,39]. In brief, GST-fusion proteins were immobilized on glutathione sepharose beads and incubated with 10 µM actin labeled with Alexa Fluor 568 in 10.9 mM creatine phosphate, 1.5 mM ATP, and 1.5 mM MgCl₂. After preincubation on ice (5 min), the reactions were initiated by a temperature shift to RT. Digital images were taken live using an AxioCam MRm CCD camera and an Axio Observer.Z1 equipped with a 20x/0.5 objective.

### Pyrene-actin polymerization quenching
Pyrene-actin polymerization assays were set up according to Ahuja et al.[9]. Pyrene-labeled rabbit skeletal muscle actin and unlabeled rabbit skeletal muscle actin were mixed and incubated overnight at 4 °C in G-buffer (5 mM Tris HCl pH 8.0, 0.2 mM CaCl₂, 0.2 mM ATP, 0.2 mM DTT). The solution was centrifuged for 1 h at 4 °C and 100000 × g to remove putative traces of insoluble actin and/or F-actin. 90 µl of the purified actin binding proteins to be tested (4.92 µM in G buffer) were incubated with 10 µl 500 mM KCl, 20 mM MgCl₂, 50 mM guanidine carbonate, 10 mM ATP, 100 mM Tris, pH 7.5 for 2 min. From this mixture, 90 µl were supplemented with 10 µl G-actin solution in a microtiterplate (black 384 well, non-binding polystyrene, F-bottom; Greiner Bio-One GmbH). Recordings of fluorescence (wavelength, 407 nm) were done for 10 min using a SpectraMax® M5 Microplate Reader and SoftMax Pro 6.3 software (excitation wavelength, 365 nm; time increment, 5 s).

Data were exported to Excel and visualized graphically using GraphPad Prism 8.4.3.

### Preparation of cell and tissue lysates and coprecipitation analyses
48 h after transfection, L929 cells were washed with PBS, harvested, and subjected to sonication (3 × 5 s) in IP buffer (10 mM HEPES, 0.1 mM MgCl₂, 1 mM EGTA, 1% (v/v) TritonX-100, 150 mM NaCl, pH 7.4 containing EDTA-free protease inhibitor Complete (Roche)) and incubated for 30 min at 4 °C[43]. Cell lysates were obtained as supernatants from

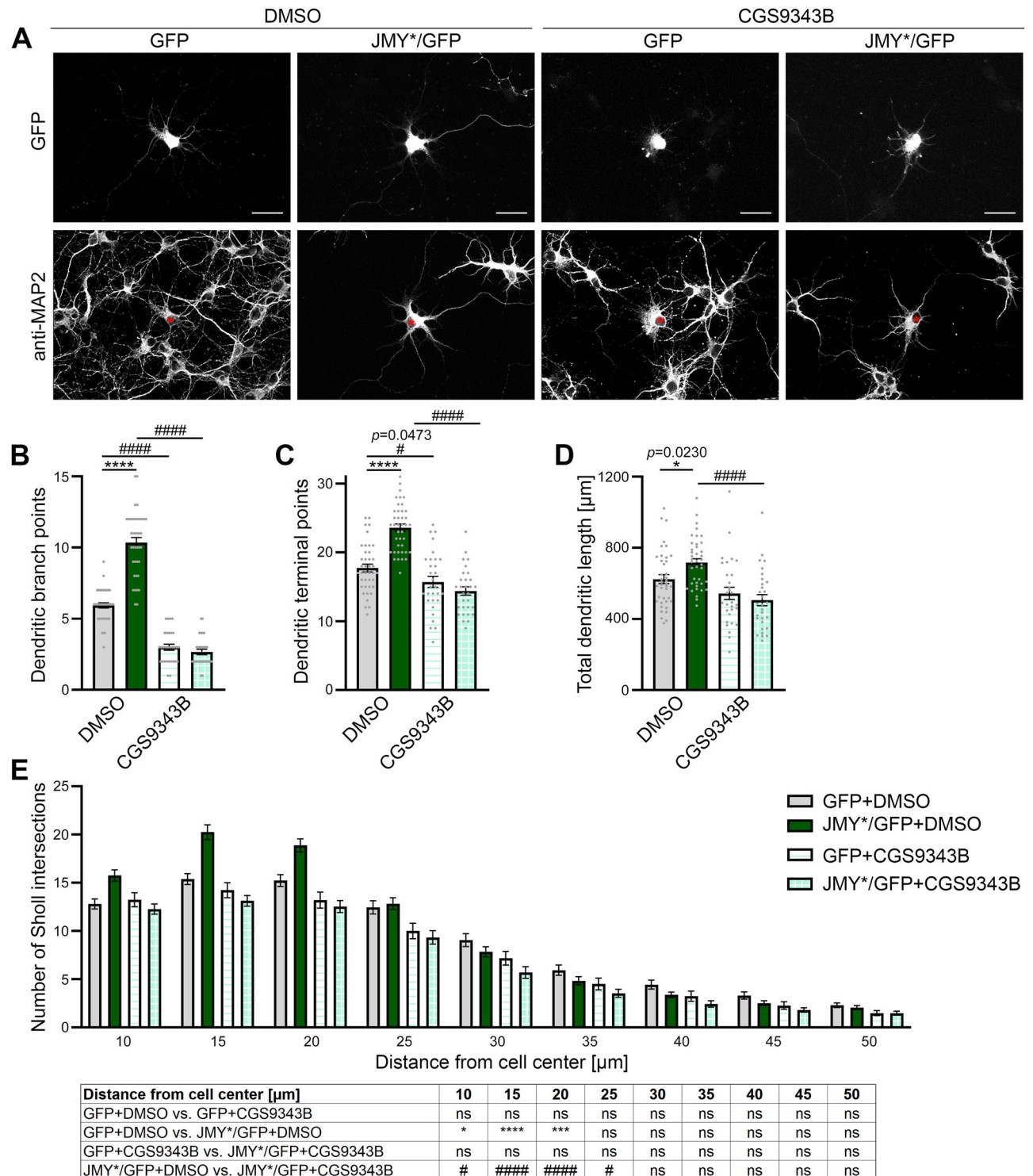

**Fig. 7 | The functions of JMY in dendritic arborization require proper Ca²⁺/CaM signaling. A** Representative MIPs of the GFP and anti-MAP2 signals of primary hippocampal neurons transfected with either GFP or JMY*/GFP at DIV4 and treated with the CaM inhibitor CGS9343B and solvent control (DMSO), respectively. Asterisks mark transfected cells. Bars 30 μm. **B–E** Imaris software-based reconstruction and quantitative analyses of the dendritic arbors of transfected neurons 30 h after transfections. Shown are determinations of dendritic branch points (**B**), dendritic terminal points (**C**), and total dendritic length (**D**) as well as Sholl analyses of dendritic arbor complexity (**E**). Note that all JMY gain-of-function phenotypes were completely abolished upon the addition of CGS9343B when compared to solvent control. Data, mean ± SEM shown as

bar/dot plots (**B–D**) and bar plots (**E**), respectively. GFP+DMSO and JMY*/GFP +DMSO (same data as in Fig. 4B–E), n = 40 each; GFP+CGS9343B and JMY*/GFP +CGS9343B n = 30 cells each from 2 independent assays and neuronal preparations. Numerical data are provided in Supplementary Data 1. Statistical significances, Two-way ANOVA, Sidak's test (**B-D**) and Three-way ANOVA with Tukey's multiple comparisons test (**E**), respectively, for the factor transfection (GFP vs. JMY*/GFP shown with *) and the factor treatment (DMSO vs. CGS9343B shown with #). *$p < 0.05$, ***$p < 0.001$, ****$p < 0.0001$, #$p < 0.05$, ####$p < 0.0001$. For exact $p$ values in **B-D** see figure panels. Note that ****$p < 0.0001$ are too small values to be reported by the software used.

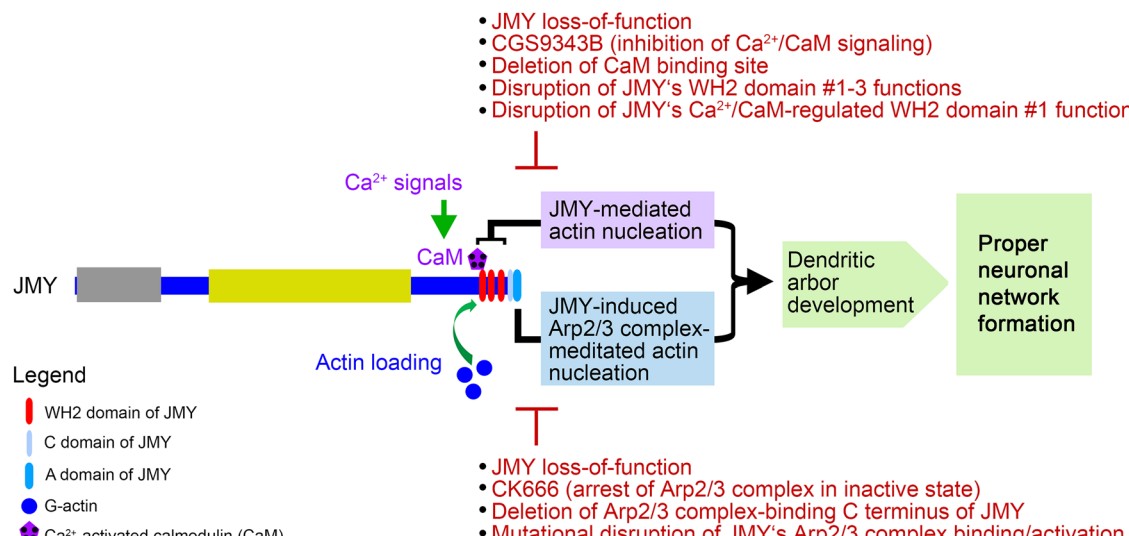

**Fig. 8 | Model summarizing the neurobiological functions of JMY, the molecular mechanisms it uses, and the regulation of the G-actin loading of JMY by Ca²⁺/CaM.**

centrifugations at $1000 \times g$ for actin precipitation and $20800 \times g$ for CaM coprecipitation assays (15 min at 4 °C).

Coprecipitation experiments with immobilized GST fusion proteins and extracts from L929 cells were essentially done as described before[35,40]. In brief, L929 cell lysates were incubated for 2 h at 4 °C with purified, recombinant GST-fusion proteins immobilized on glutathione sepharose beads. The samples were washed several times with IP buffer. Subsequently, bound protein complexes were eluted either with 20 mM glutathione, 120 mM NaCl, and 50 mM Tris-HCl, pH 8.0 (30 min at 4 °C) or by incubating the samples for 5 min at 95 °C in 2x SDS sample buffer including 4 M urea.

For coprecipitations with immobilized CaM, lysates of L929 cells expressing different GFP-JMY fusion proteins were prepared essentially as described before[8,16] in an EGTA-free IP buffer. Cell lysates were supplemented with either 1 mM EGTA or with 2 and 500 μM CaCl₂, respectively. After incubation with 40 μl CaM-sepharose 4B for 3 h at 4 °C and washing, bound proteins were isolated by incubation at 95 °C for 5 min in 2x SDS sample buffer including 4 M urea. Lysates, supernatants, and eluates were analyzed by immunoblotting using anti-GST and anti-GFP antibodies, respectively.

Rat brain lysates were prepared in homogenization buffer (10 mM HEPES, 1 mM EGTA, 150 mM NaCl, 0.1 mM MgCl₂, pH 7.4) complemented with protease inhibitors, using an ultra-turrax. Triton X-100 was added in a final concentration of 1% (v/v). The solution was incubated by rotating it 15 min at 4 °C. The supernatant of a subsequent centrifugation at $100000 \times g$ and 4 °C for 1 h was collected for subsequent experiments. The protein concentration was determined, and for coprecipitation analyses, a lysate volume corresponding to a content of 1000 μg protein was added to each sample of immobilized GST fusion protein.

Immunoblotting analyses were performed with anti-β-actin and anti-GST antibodies.

**Coimmunoprecipitation analyses**
Coimmunoprecipitations of endogenous JMY and CaM were performed using Hela cell (RRID:CVCL_0030) lysates in 10 mM HEPES, 0.1 mM MgCl₂, 1 mM EGTA, 1% (v/v) TritonX-100, 30 mM NaCl, pH 7.4, containing EDTA-free protease inhibitor Complete and mouse anti-CaM (G-3) antibodies according to coimmunoprecipitation procedures described previously[8,35].

Coimmunoprecipitation analyses of endogenous actin together with GFP and GFP-tagged JMY deletion mutants were done according to Hou et al.[8] with slight modifications. Lysate from L929 cells transfected with GFP-fusion constructs was incubated with rabbit anti-GFP antibody or non-immune rabbit IgG as a negative control for 1 h at 4 °C.

Subsequently, 35 μl protein A/G PLUS-agarose matrix (Santa Cruz) was added to the lysate/antibody mixture. After incubation for 3 h at 4 °C, the samples were centrifuged for 1 min at $20800 \times g$. After three washing steps, elutions of the coimmunoprecipitates from the protein A/G agarose matrix were done for 5 min at 100 °C in a mix of 8 M urea and 4x SDS-sample buffer. Ca²⁺ dependencies were evaluated by treating the lysate/antibody-mixture with either final concentrations of 2 μM Ca²⁺ or 1 mM EGTA, respectively.

The samples were immunoblotted and analyzed using fluorescently labeled secondary antibodies and a LI-COR Odyssey System. Quantiative analyses were done using Image Studio Lite System 5.2 software (LI-COR). The anti-actin signal of each coimmunoprecipitation was expressed as ratio per immunoprecipitated GFP-JMY fusion protein (anti-GFP signal). Conditions with EGTA (no-Ca²⁺ condition) were considered as basic levels and set to 100%. Calcium-mediated changes were expressed as percent deviation from the corresponding values of the no-Ca²⁺ conditions in the assay.

For coimmunoprecipitations mimicking transient Ca²⁺-conditions, 1 mM EGTA (final concentration) was added for 1 h after the samples had previously been treated with 2 μM Ca²⁺ (final concentration) for 1 h, while the respective no-Ca²⁺ and Ca²⁺ condition samples were incubated in their respective continuous condition for 2 h. Antibody-associated protein complexes were isolated with protein A/G agarose matrix, washed with a buffer with the respective Ca²⁺ concentrations, eluted, and analyzed and quantitatively examined as described above.

**Precipitations of endogenous JMY from HeLa cells lysate**
Precipitates of endogenous JMY with CaM immobilized on Sepharose 4B were obtained by incubating HeLa cell lysates prepared in EGTA-free IP buffer as described above in the presence of 2 and 50 μM CaCl₂, respectively, or in the absence of calcium (1 mM EGTA added).

Bound proteins were eluted by incubating the CaM-sepharose matrix with 4× SDS sample buffer for 5 min at 95 °C. Lysates and eluates were analyzed by anti-JMY immunoblotting.

**Cell culture, transfection, and immunostaining**
Culturing of L929 cells (DSMZ no: ACC 2), HeLa cells (RRID:CVCL_0030), HEK293 cells, and COS-7 cells (RRID:CVCL_0224), phalloidin-stainings and immunolabelings were essentially as described[44]. L929, HEK293, and COS-7 cells were transfected by using TurboFect (Thermo Fisher Scientific).

For subcellular recruitment assays, mitochondria of HEK293 cells were additionally stained with 0.2 μM MitoTracker Deep Red 633 in medium at 37 °C for 1 h before the cells were fixed with 4% (w/v) PFA in PBS for 7 min.

3D-membrane ruffling of phalloidin-stained COS-7 cells transfected with GFP and JMY*/GFP, respectively, was essentially determined as described[9,16].

Cultures of primary rat hippocampal neurons (see below) for immunofluorescence analyses were prepared, maintained, and transfected (at DIV4) as described[35,39]. Fixation was done at DIV6 in 4% (w/v) PFA in PBS pH 7.4 at RT for 4–6 min.

Permeabilization and blocking were done with 10% (v/v) horse serum, 5% (w/v) BSA in PBS with 0.2% (v/v) Triton X-100. Antibody incubations were done in the same buffer without Triton X-100 according to Haag et al.[45].

Images were recorded by using an AxioObserver.Z1 microscope (Zeiss) equipped with an ApoTome, with Plan-Apochromat 100×/1.4, 63×/1.4, 40×/1.3, and 20×/0.5 objectives and an AxioCam MRm CCD camera (Zeiss). Digital images were recorded by ZEN2012. Image processing was done by Adobe Photoshop.

Neuronal cultures for Western blot analyses were prepared from cortices or hippocampal preparations from E18 rat pups, essentially as described (cultures of rat cortical neuron[46]; cultures of hippocampal rat neurons[47]). The cells were lyzed at DIV14 in IP buffer. After centrifugation, the lysates of the cultured neuronal cells were subjected to immunoblotting.

## Culturing, transfection, and immunostaining of primary rat neurons

Rats (Crl:WI; Charles River) were used for primary cell culture preparations from brain tissue. Animal housing and breeding were done by the animal facility of University Hospital Jena in strict compliance with the EU directives 86/609/EWG and 2007/526/EG guidelines for animal experiments. All procedures were approved by the local government (Thüringer Landesamt, Bad Langensalza, Germany, and Landesverwaltungsamt; allowance for primary hippocampal culture preparations from rat embryos (UKJ-24-005)). Rat hippocampal neurons were prepared and cultured in 24-well plates until day in vitro (DIV) 4 as described in refs. 10, 46. The neurons were transfected with 1 µl Lipofectamine 2000 and 1 µg DNA per well in antibiotic-free medium. After 4 h, the transfection medium was replaced by conditioned medium. About 30 h after transfection, neurons were fixed with 4% (w/v) PFA for 5 min.

For inhibitor experiments, the cultured primary neurons were cultured with CK666 (stock solution 100 mM in DMSO) and the CaM inhibitor CGS9343B (1,3-dihydro-1-[1-[[(4-methyl-4H,6H-pyrrolo[1,2-a][4,1]-benzoxazepin-4-yl)methyl]-4-iperidinyl]-2H-benzimidazol-2-one(1:1) maleate; Tocris) (stock solution 20 mM in DMSO), respectively, for 6 h before fixation at in total 30 h post-transfection. Final concentrations of 50 µM CK666 and 10 µM of CGS9343B were applied in final concentrations of 0.05% (v/v) DMSO. A similar concentration of DMSO was used as solvent control.

After fixation, neurons were immunostained[45,48,49]. In brief, neurons were incubated in blocking solution consisting of 10% (v/v) horse serum, 5% (w/v) bovine serum albumin and 0.2% (v/v) Triton X-100 in PBS (1 h at RT) and then immunostained with primary antibodies and with primary antibodies and fluorescently labeled phalloidin, respectively, washed three times with blocking solution and incubated with secondary antibodies and DAPI. After washing, the coverslips were mounted onto glass slides using Moviol.

Quantitative analyses of anti-JMY immunostainings for validation of JMY RNAi were conducted in primary hippocampal neurons transfected at DIV4, fixed 30 h later, and anti-JMY immunostained following the procedure described above.

Anti-JMY immunosignals of neurons transfected with JMY RNAi and with scrambled RNAi, respectively, were measured using sum intensity projections. Neurons transfected with untagged JMY* served as positive controls for successful anti-JMY immunodetections. A circular ROI was placed in the cytoplasm of the RNAi-expressing neuron and in the cytoplasm of a neighbored untransfected neuron, which served as intrinsic control for determining the endogenous JMY levels, using ImageJ software. The anti-JMY immunosignals of RNAi-expressing cells were expressed as ratios per intrinsic control cell intensities and compared to ratios obtained for neurons expressing scrambled RNAi.

## Imaging and quantitative assessments of neuronal morphology

Transfected neurons from 2 to 5 independent assays were imaged by systematic sweeps across the coverslips using a Zeiss Axio Observer Z1 microscope/Apotome (objectives, Plan-Apochromat 100×/1.4, 63×/1.4, 40×/1.3, and 20×/0.5) and an AxioCam MRm CCD camera (Zeiss). Digital images were recorded by ZEN2012 (RRID SCR_013672). Image processing was done by Adobe Photoshop (RRID SCR_014199).

Morphometric measurements were based on anti-MAP2 immunosignals to identify dendrites of neurons. The successful expression and the localization of untagged JMY* and mutants thereof were addressed by parallel analyses of anti-JMY immunostainings. Neuronal morphologies were reconstructed in 3D using Apotome image stacks processed by Imaris 8.4.0 software (RRID SCR_007366) using the following settings: start seed point 15 µm, seed points 2 µm, and minimum size of considered segments 10 µm.

The number of dendritic branching points, terminal points, total dendritic tree length, and Sholl intersections[50] were analyzed as described previously in ref. 9. Additionally, branch depth levels (with level 1 representing primary dendrites) were determined as described recently[51] using Imaris 8.4.0. For a schematic illustration of the dendritic evaluation parameters, see Supplementary Fig. 1E. For examples of Imaris-based morphology tracings, see the 2D representations of 3D Imaris *filaments* with their branch and terminal points shown in Supplementary Figs. 1D and 2A.

## Statistics and reproducibility

No explicit power analyses were used to compute and predefine required sample sizes. Instead, all neuronal analyses were conducted by systematic sampling of transfected cells across coverslips to avoid any bias. Morphometric analyses were then conducted by using Imaris software.

Outliers or strongly scattering data most likely reflect biological variance. Therefore, outliers were not excluded from the data. All $n$ numbers reported in the manuscript are numbers of independent biological samples (i.e., neurons) or biochemical assays, as additional replicates to minimize measurement errors were not required because the technical errors were small in relation to the biological/biochemical variances.

All quantitative data shown represent mean ± SEM. Tests for normal data distribution and statistical significance analyses were done by using GraphPad Prism 9 software (SCR_002798).

Normal data distribution was tested using the Shapiro–Wilk Test, the Anderson–Darling Test, and the Kolomogorov–Smirnov Test.

In case of a normal distribution and comparison of two conditions, an unpaired Student's $t$-test or a Welch's $t$-test was performed. When more than two conditions were compared, a One-way analysis of variance (ANOVA) with a Tukey's post-test was conducted.

If the Shapiro–Wilk Test, the Anderson–Darling Test, or the Kolomogorov–Smirnov Test proposed non-normal distribution, a Mann–Whitney $U$-test was performed for the comparison of two conditions, and a Kruskal–Wallis test with Dunn's multiple comparison test was conducted for the comparison of more than two conditions.

Two-factor analyses (e.g., Sholl analyses) were examined for statistical significances using Two-way ANOVA with a Bonferroni's multiple comparison test or Sidak's test. When three factors were compared, Three-way ANOVA with Tukey's post-test was used.

Statistical significance was marked by $*p < 0.05$, $**p < 0.01$, $***p < 0.001$, $****p < 0.0001$ throughout. Statistical significances of additional factors analyzed were marked by $\#p < 0.05$, $\#\#p < 0.01$, $\#\#\#p < 0.001$ and $\#\#\#\#p < 0.0001$ correspondingly. The exact $p$ values are reported directly in the most of the figure panels. Note that $****p < 0.0001$ are too small values to be reported by the software used (Prism).

## Reporting summary

Further information on research design is available in the Nature Portfolio Reporting Summary linked to this article.

## Data availability

The authors declare that all data supporting the findings of this study are available within the paper and its supplementary information files.

## Code availability

The software used was either commercial or freely available and is listed and specified in the Materials and Methods section.

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

## Acknowledgements
We thank K. Gluth, B. Schade, A. Kreusch, S. Berr, M. Öhler, and M. Röder for excellent technical assistance and the students L. Kloß, M. Kaufmann, and Y. Zhang for help with some initial tool constructions and validations. We are furthermore very grateful to M. D. Welch (University of California, Berkeley) for his generous gift of a JMY-encoding plasmid (pLAP-JMY), which served as template for our JMY tool constructions. This work was supported by D.F.G. grants QU116/10-1 to B.Q. and KE685/7-1 to MMK and by the Interdisciplinary Center of Clinical Research of the Medical Faculty Jena (IZKF, MSP 03) to M.I.-S.

## Author contributions
M.K. conducted the majority of experiments shown. M.I.-S. also conducted experiments and interpreted data. A.-L.Z. provided valuable initial observations and insights. M.K., M.M.K, and M.I.-S. visualized data. B.Q., M.M.K., and M.I.-S. conceived the project, designed experiments, interpreted data, provided scientific supervision and funding, and wrote the manuscript.

## Funding

## Competing interests
The authors declare no competing interests.
