## [Transparent Peer Review file · Communications Biology]

JMY powers dendritogenesis and is regulated by CaM revealing a general, critical principle in neuromorphogenesis

Corresponding Author: Dr Michael Kessels

Version 0:

Reviewer comments:

Reviewer #1

(Remarks to the Author)

The manuscript by Kühne et al. provides significant insights into the role of the protein Junction-mediating and regulatory protein (JMY), an actin nucleator, in dendrite development. Although JMY is highly expressed in brain, its role in neurodevelopment is not well understood. Here they show that JMY is required to promote dendrite arborization in cultured neurons and show that JMY actin nucleation activity is regulated by Ca²⁺/calmodulin association. Additionally, by using structure function analysis of JMY they determine the requirements of specific domains within JMY for dendrite branching. Overall, this is an interesting study and would recommend for publication in Communications Biology.

Specific comments:

Figure 1: The description of experimental design in panel A needs clarity, especially regarding the IP conditions (which protein is being immunoprecipitated etc). A schematic would help this.

For panel B, please include quantification to substantiate claims about the dependency on Ca²⁺ or remove strongly from the statement "was strongly Ca²⁺-dependent".

In panel D, a rationale for the use of mitochondria-targeted CaM over a simpler construct would be insightful.

For panels G, H-J, and K-L, including more detailed annotations, zoomed views of individual cells, and clearer descriptions in figure legends would help in understanding the results better.

Figure 2: One of the most striking results in this figure is the hyperbranching seen with JMY's re-expression "rescue" experiment. Although all the various dendritic parameters measured are significantly higher than the scrambled RNAi condition presumably due to overexpression, and so the re-expression of the WT is not technically a full rescue. The fact that the JMY "rescue" does not fully rescue the phenotype or causes overexpression phenotypes needs to be discussed.

Figure 3 and Figure 5: These figures suggest that truncations and mutations might affect protein expression or functionality. There is no quantification of the expression levels and some of the truncations and mutants could be due to expression level or stability. It would be beneficial to validate expression levels or localization of these JMY variants to confirm that differences are not due to variable expression or stability. Minimally this needs to be acknowledged in the text.

In Figure 5, the specific region truncated for JMY*ΔCaM not clear. A figure with the domains and the residues corresponding to the regions would be helpful.

Figure 7: Although the CaM inhibition suppresses the effects of JMY overexpression, it could impact dendritic morphology independently of JMY as it can affect other pathways. It is important at least mention this possibility that CAM could influence dendritic branching mechanisms through pathways distinct from those involving JMY.

Typo:

Line 154 , 277, "analyses"

Line 284 "deficient"

Reviewer #2

(Remarks to the Author)

This manuscript by Kuhne et al. examines the regulation of neuronal morphogenesis by calcium, calmodulin, and the actin nucleation factor JMY. Using several immunoprecipitation strategies along with cell biological assays of dendritogenesis that combine RNAi of JMY with plasmid complementation using JMY mutants, the authors found that one of the actin-binding motifs (WH2#1) in the C-terminal region of JMY doubles as a calmodulin-binding sequence. This WH2#1 is crucial for JMY's to promote dendritogenesis. The Arp2/3 complex binding CA sequence at the C-terminus of JMY is also critical. Overall, this is an interesting paper packed with data. However, a few conclusions are incompletely supported by the images that were provided (points 1-3), and it is unclear if the authors have a model for how actin-, calmodulin-, and Arp2/3-binding are coordinated (points 4-6).

Specific Points

1. The endogenous localization of JMY in neurons in Figure 1E is unclear. Most JMY appears to be in the nucleus, and the so-called "dendritic localization pattern" of JMY and colocalization with MAP2 (lines 95-96) is weak and not very convincing. Do the authors have better images to support their conclusions that JMY is in dendrites? Do they have images of neurons stained for other WASP-family proteins to compare as negative controls?
2. Several of the measurements of neuronal morphogenesis are unclear to non-specialist readers. For example, in Figure 1 G,H,I,J, it would be very helpful if the authors provided magnified versions of individual neurons to explicitly highlight the number of dendritic branch/terminal points and length. These could be incorporated into the main figure or at the very least in the supplement. The methods in lines 695-697 should also be expanded beyond the citation of previous papers.
3. Similar to point #2, in Figure 2A it is not apparent to the reader that the RNA#1-expressing cell in the middle of the image is different from any of the surrounding nontransfected cells. Moreover, in the accompanying Figure S2, it is unclear if JMY is depleted in the transfected cell, as MAP2 fluorescence also looks lower in the transfected cell. Can the authors estimate the % JMY knockdown by fluorescence intensity normalized to MAP2? Is the intensity or localization of F-actin or Arp2/3 complex staining altered in any way in JMY-depleted cells? The Figure 2 title stating that JMY is "required" for arborization is too strong, given that the most extreme phenotype in the figure (dendritic branch points) is only a ~2-fold reduction from the control.
4. Figure 6I seems like a key panel for examining the direct binding of calmodulin and actin to JMY, and for examining potential competition between calmodulin and actin. Can the authors gain more quantitative information about affinity by using different concentrations of the proteins? Perhaps more importantly, does calcium alone (in the absence of calmodulin) affect actin binding to JMY in these assays?
5. Similar to point #4, the conclusions of how calmodulin, actin, and Arp2/3 binding are coordinated are ambiguous. While point #4 can address the actin-calmodulin question, do the authors know whether WH2#1 combines with the CA sequence to promote Arp2/3 activation? The original results of Zuchero et al. imply that WH2#1-3 can nucleate actin directly, while WH2#3 can serve an Arp2/3-activating function with CA in vitro, but it is unclear if WH2#1 is functioning in direct nucleation or Arp2/3 activation in the current paper. What do the authors think is happening in their neuronal experimental system?
6. The cartoon in Figure 8 summarizes the authors' observations but does not address how the C-terminal functions of JMY are coordinated in the cell. Any advances related to points #4-5 would provide a clearer take-home message for the paper. Is calmodulin important for JMY localization or activation, and how/when might calcium fluctuations change the function of JMY? Insight (and some speculation) into the spatiotemporal regulation of JMY would be welcomed.

Minor

1. Lines 69-70, what makes this mechanism "powerful"?
2. Line 90 & 272, cell culture should not be classified as "in vivo" - a term which more often refers to experiments within a live animal.
3. Line 113, what are Sholl analyses? Some contextualization for non-specialist readers (as in major point #2 above) would be appropriate here.
4. The results in Figure 3 are not surprising, as (to my knowledge) no WASP-family member can be split in half and retain normal function in cells. If the paper needs focusing/shortening, this figure could be moved to the supplement.
5. Lines 185-187, have any previous studies examined the impact of CK666 treatment on neuronal morphogenesis/dendritogenesis? If so, they should be mentioned/cited here.
6. Line 382, any in-text referral to unpublished observations should be removed or supported by data added to the supplement.

Reviewer #3

(Remarks to the Author)

This paper identified the WH2 domain-containing actin nucleator protein JMY as a novel, Ca²⁺-dependent direct binding partner of Calmodulin, and demonstrated that the complex of JMY and CaM regulates dendritic development. Through loss-

of-function studies mediated by shRNA knockdown of JMY protein levels, coupled with overexpression of domain deletion constructs, the authors determined that JMY is necessary for normal dendritic branch development, and this function is dependent on both the N-terminal Arp2/3-binding domains and C-terminal G-actin-binding domain. Functional experiments further elucidated the domain in JMY that binds to CaM, which partially encompasses the first WH2 domain, and that JMY's first WH2 domain is the primary domain contributing in G-actin association. Finally, the association of JMY with actin through its first WH2 domain was shown to be regulated by Ca²⁺-activated CaM, with an increase in CaM binding resulting in a decrease in JMY actin binding. The following points need to be addressed.

Major points:

1. This study relies on the use of shRNA-mediated knockdown of JMY to determine the role of this protein in dendritic development. However, validation is needed to support the knockdown of JMY at the protein level. Fig. S2A shows immunofluorescence staining against JMY in a single neuron expressing RNAi#1. Given that the findings of this paper are dependent on knockdown of JMY, more validation is necessary for the shRNA knockdown. Ideally, this should be demonstrated by quantitative Western blotting against JMY in shControl or shJMY expressing neurons. Given the low transfection efficiency of primary neurons, immunofluorescence staining of neurons expressing control or JMY shRNA would also be acceptable, but a larger n than one neuron is necessary, and shControl-expressing neurons should also be included in this analysis.
2. The neuronal images presented in this paper are shown with MAP2 staining as a marker of dendrites, and transfected cells are marked with a star. However, data showing multiple channel images with both GFP and MAP2 have not been included for any of the neuronal images, and should be included for transparency. This would allow comparison of transfected and non-transfected neighboring neurons.
3. In Figure 1C, CaM was immunoprecipitated from HeLa cell lysate, and JMY co-immunoprecipitation was observed only in the anti-CaM sample. However, there are bands present in the IgG control sample for CaM, making interpretation of this experiment challenging. A representative blot without bands in the IgG control lane for CaM would lend credence to the conclusions of this figure.
4. Again in Figure 4, why not show the GFP-construct instead of indicating cells with an asterisk? This is important to show the localization and level of overexpression of the GFP-constructs
5. Please show the localization of the different GFP-tagged deletion constructs to show how different deletion domain constructs localize.

Minor points:

1. In figure 1D, the figure legend describes the scale bar as being 10 um in both images, but the cells in the lower image appear much larger than the cells in the top image. Representative images showing the same zoom for both conditions would strengthen the conclusion of this figure.
2. JMY overexpression, either alone or as a rescue for JMY shRNA, is used in a number of figures in this paper. However, staining against JMY in these cells has not been performed. Inclusion of immunofluorescence staining of JMY would be informative about the degree of overexpression of JMY in these cells, especially in comparison to the surrounding cells with endogenous JMY expression.
3. Minor grammatical or clarity issues were identified in the following lines:
 - a. Line 96: exiting is used rather than existing
 - b. Line 208: yielded is misspelled
 - c. Line 277: a word may be missing in this sentence after similarity: "The similarity _____ the actin coimmunoprecipitations levels by JMY CT and by JMY WH2#1 suggested that the actin association seen in coimmunoprecipitations mostly reflected the contribution of the first WH2 domain (Fig. 6C,D)."
 - d. Line 294: Controlled is misspelled
 - e. Line 309: The underlined phrase is ambiguous: more than 80% of...? Clarification is needed here. "With more than 80%, the determined decline was very severe in both cases (Fig. 6L)."

Version 1:

Reviewer comments:

Reviewer #1

(Remarks to the Author)

The authors have addressed all my comments and improved the manuscript significantly. I would recommend publication of the manuscript.

Reviewer #2

(Remarks to the Author)

A lot of work went into this paper. Well done.

Reviewer #3

(Remarks to the Author)

My concerns have been addressed by the revisions presented by the authors. I recommend acceptance of the manuscript for

publication.

Detailed Point-to-point responses to the reviewer comments

Reviewer #1 (Remarks to the Author):

The manuscript by Kühne et al. provides significant insights into the role of the protein Junction-mediating and regulatory protein (JMY), an actin nucleator, in dendrite development. Although JMY is highly expressed in brain, its role in neurodevelopment is not well understood. Here they show that JMY is required to promote dendrite arborization in cultured neurons and show that JMY actin nucleation activity is regulated by Ca²⁺/calmodulin association. Additionally, by using structure function analysis of JMY they determine the requirements of specific domains within JMY for dendrite branching. Overall, this is an interesting study and would recommend for publication in Communications Biology.

We thank the reviewer for his/her positive assessment of our study and its impact.

Specific comments:

Figure 1: The description of experimental design in panel A needs clarity, especially regarding the IP conditions (which protein is being immunoprecipitated etc). A schematic would help this.

We thank the reviewer for the note that the experiments represented by Figure 1A may have been hard to understand. **We made three improvements** in order to have readers understand better that Fig. 1A is a coprecipitation with immobilized calmodulin matrix (not an immunoprecipitation) and how these experiments are done and have to be interpreted. **First, we added a schematic representation of such experiments to Figure S1 (revised Fig. S1A). Second, the details of the experiments (including that immobilized CaM was used) are now more explicitly described in the legend of the revised Fig. 1. Finally, third, the revised Fig. 1A itself now explicitly states that immobilized CaM has been used for the coprecipitations.**

We hope that the reviewer will be fine with these changes to improve the manuscript.

For panel B, please include quantification to substantiate claims about the dependency on Ca²⁺ or remove strongly from the statement “was strongly Ca²⁺-dependent”.

We quantified the dependency of the JMY interaction with calmodulin on Ca²⁺, as suggested by the reviewer. These data sets are now included in the revised Figure 1B.

We obtained a very high increase of binding upon both 2 μM and 50 μM Ca²⁺ when compared to conditions without Ca²⁺ (EGTA). Without Ca²⁺, calmodulin virtually did not associate with JMY. Therefore, the binding upon presence of Ca²⁺ was more than an order of magnitude higher than control and also statistically significant in quantitative Western blot analyses (**revised Fig. 1B**). Therefore, we described the CaM/JMY complex formation as strongly Ca²⁺-dependent. The quantitative representation of the data firmly underscores this. We thank the reviewer for his/her suggestion.

In panel D, a rationale for the use of mitochondria-targeted CaM over a simpler construct would be insightful.

We thank the reviewer for this note. The description of this experiment may indeed have been too short to be easily understood by readers not familiar with this type of assay, as such *in vivo*-reconstitutions are rarely successfully conducted and described in literature.

The rationale behind such experiments is now described in more detail in the revised version of the manuscript.

A successful and specific reconstitution of protein complexes at some pre-defined place (here the outer mitochondrial membrane surface presenting a mito-Cherry-CaM fusion protein towards the cytosol)

strongly supports the *in vivo* relevance of the newly identified protein interaction. By relocalization of the to-be-recruited binding partner, such protein complexes can directly be proven and visualized. The specificity and the CaM dependence of the obtained relocalization of JMY to mitochondrial surfaces is proven by the fact that the anti-JMY immunostaining does not show any mitochondrial accumulation in untransfected cells or in cells merely transfected with mito-Cherry instead of mito-Cherry-CaM. The fact that the protein complex formations were visualized in cells hereby firmly excluded that the protein interaction observed in e.g. coprecipitation and coimmunoprecipitation experiments may represent putative post-solubilization artifacts. Instead, such experiments firmly demonstrate protein complex formations in intact cells.

For panels G, H-J, and K-L, including more detailed annotations, zoomed views of individual cells, and clearer descriptions in figure legends would help in understanding the results better.

We thank the reviewer for the note that the quantitative analyses of neuronal development may need more explanations by further information/visualizations. **The revised manuscript now includes both more raw data information and more information on the evaluation:**

We included a schematic drawing of a developing neuron to illustrate the evaluated branch points, terminal points, dendritic branch depth and Sholl analyses (Supplementary Figure 1E). We expanded Supplementary Figure 1D by adding the corresponding GFP channels of the neurons shown in Fig. 1G (GFP is used as a reporter for JMY expression; it also highlights the overall morphology of the transfected neuron but note that the endogenous anti-MAP2 immunostainings are used to track dendritic arbors of transfected cells) and enlarged these pictures, as the reviewer suggested.

In addition, we added pictures of the reconstruction of dendritic arborization by Imaris software (Imaris *filament*) to the revised Supplementary Figure 1D. We did the same for the example images shown in Fig. 2 (which furthermore also was complemented with the corresponding GFP channels). These Imaris reconstruction images also contain dendritic branch point and terminal point markings by Imaris (same color as in our scheme), which become visible when one zooms in. We added this information to the legends of Fig. S1 and Fig. 2.

The GFP pictures of the neurons shown in Fig. 3 and 6 are now shown in the corresponding revised Supplementary Figures.

In the case of Fig. 4A, we integrated these additional images into the main figure (revised Fig. 4A). The images of additional mutational analysis shown in Fig. 4 are shown in the Supplementary Figure 4.

The Supplementary Figure S5 also has been enlarged by inclusion of the corresponding GFP-images.

Figure 2: One of the most striking results in this figure is the hyperbranching seen with JMY's re-expression "rescue" experiment. Although all the various dendritic parameters measured are significantly higher than the scrambled RNAi condition presumably due to overexpression, and so the re-expression of the WT is not technically a full rescue. The fact that the JMY "rescue" does not fully rescue the phenotype or causes overexpression phenotypes needs to be discussed.

The reviewer has observed this correctly. The RNAi-insensitive JMY mutant (JMY*) is introduced by transfection and the resulting expression of JMY* thus of course exceeds endogenous levels (also see **Reviewer Fig. 1** below). Therefore, the identified JMY loss-of-function phenotypes are not only completely rescued but the rescue overshoots to some extent (please also compare JMY gain-of-function analyses in Fig. 1). In some cases the RNAi/JMY* condition even displays some overexpression effects when compared to control. In other cases, JMY loss-of-function phenotypes were fully rescued by JMY* reexpression but the rescue experiment did not lead to a gross phenotypical overshoot as no statistical significances to control values were observed (dendritic length, branch depth level 4, Sholl

intersections at 25, 30, 35 μm).

The fact that the rescue attempt is not solely compensating for all JMY loss-of-function phenotypes but actually in part already results in JMY gain-of-function phenotypes is now explicitly mentioned in the revised manuscript.

It should be noted, however, that this issue is not of relevance for the final conclusion, that the JMY* reexpression is able to compensate for JMY loss-of-function and the JMY RNAi effects thus are specific.

Figure 3 and Figure 5: These figures suggest that truncations and mutations might affect protein expression or functionality. There is no quantification of the expression levels and some of the truncations and mutants could be due to expression level or stability. It would be beneficial to validate expression levels or localization of these JMY variants to confirm that differences are not due to variable expression or stability. Minimally this needs to be acknowledged in the text.

The concern of the reviewer is justified. If mutants analyzed in rescue experiments would fail to reach at least the expression levels of endogenous protein, a not (fully) successful rescue could theoretically simply reflect a lack of the mutant rather than its hampered functionality. It therefore always needs to be checked that mutants analyzed *in vivo* express sufficiently well.

We did both, i) we analyzed by quantitative immunofluorescence analyses in developing neurons, whether the mutants express well above endogenous background and to somewhat equal levels (**see Reviewer Figure below**) and ii) furthermore evaluated their localizations. **All mutants - even the ones carrying larger deletions - are showing neuronal expression levels in the same order of magnitude as the wild-type JMY***. Also when compared to each other, there were no statistical differences among the different JMY* mutants tested. All mutants thus clearly were reexpressed in excess over the endogenous JMY (**see Reviewer Fig. 1 below**).

We therefore decided to include images of all the mutants directly into the manuscript. The images we added to Fig. 3, Fig. 4, Fig. 5 and Fig. S6 (**see inserted panels Fig. 3F, Fig. 4J, Fig. 5D,K; Fig. S6D**) clearly make the points that **i) re-expressed JMY* and all mutants thereof are available in excess** and that **ii) no gross differences are detected in neither expression levels nor localization when compared to wild-type JMY* reexpression** (in this context it may be noteworthy that the neurons expressing all mutants have far less elaborate dendritic trees. Thus, the JMY* presence in the dendritic periphery may appear to be reduced in some cases but in fact the JMY* presence in the cellular periphery is simply reduced in accordance with the dendritic arbor).

Together these new evaluations clearly demonstrate that the reason for JMY* mutants failing to rescue clearly was not their insufficient supply or lack of stability but their dysfunctionality.

Reviewer #1 Reviewer Figure 1. JMY* and all of its mutants analyzed throughout the study are expressed and furthermore show a clear excess over endogenous JMY levels

Quantitative analyses of anti-JMY immunofluorescence signals of primary hippocampal neurons, that were transfected with JMY* and the indicated JMY* mutants. Data, mean±SEM visualized as bar/dot plots. Statistical analyses One-way ANOVA/Tukey's, not statistically different (n.s.) when compared to WT JMY* and when mutants were compared among each other.

Furthermore, as the reviewer suggested, the Material and Method section of the revised manuscript now points out that we of course checked the expression levels and localization of JMY* mutants.

Together, the improvements in the revised manuscript should make it obvious for the reader that the reason for JMY* mutants failing to rescue the JMY loss-of-function phenotypes clearly was not their insufficient supply or stability but clearly their dysfunctionality.

In Figure 5, the specific region truncated for JMY*ΔCaM not clear. A figure with the domains and the residues corresponding to the regions would be helpful.

We thank the reviewer for his/her suggestion. **We added a schematic representation of JMY with its domains and the ΔNT and ΔCT deletions mutants constructed and analyzed in our study to the revised Fig. 3, a scheme of the ΔCA mutant to the revised Fig. 4 and a scheme of the ΔCaM mutant to the revised Fig. 5 and also included the exact amino acid numbers into these schemes.** All mutants had been described in Materials and methods section. **The Results section of the revised manuscript now also contains a better description of the mutants.**

We hope that with the changes made to Fig. 3, Fig. 4 and Fig. 5 as well as with the repetitions of the exact amino acids of the deletion mutants in the Results text, these informations are now better accessible for the readers, as they do not have to get the details from the Material and Method section anymore.

Figure 7: Although the CaM inhibition suppresses the effects of JMY overexpression, it could impact dendritic morphology independently of JMY as it can affect other pathways. It is important at least mention this possibility that CAM could influence dendritic branching mechanisms through pathways

distinct from those involving JMY.

We thank the reviewer for the hint that the interpretation may need to be even more explicit to have readers understanding that our analyses revealed i) a full suppression of all JMY-induced gain-of-function effects on dendritic arbor formation and that ii) the inhibition by CGS9343B does not merely reflect some suppression of putative parallel pathways and mechanisms unrelated to JMY. As we explicitly studied JMY-mediated dendritic branch induction, other CaM effects unrelated to those of JMY of course are well possible and these also are discussed in the manuscript.

We would also like to refer the reviewer to the improved description in the Results section. As in the previous manuscript version, the revised manuscript first describes that CGS9343B has some effect on dendritic arbor formation in control cells (these effects may reflect a set of currently mostly unknown effectors being addressed), then describes the suppression of the specifically JMY gain-of-function phenotypes and then finally compares the two conditions (“The observed suppression of JMY-mediated dendritic arborization was so strong that dendritic branch points, terminal points and total dendritic length of CaM inhibitor CGS9343B-treated control neurons expressing solely GFP and of neurons expressing JMY*/GFP were at similar levels (Fig. 7B-D).” **The discussion and conclusion also was improved.** **We hope that the reviewer will be content with the more extensive and explicit version in the revised manuscript.**

Typo:

Line 154 , 277, “analyes”

We thank the reviewer for drawing our attention to this typo (we actually found it a second type in the manuscript). **We corrected this in the revised manuscript.**

Line 284 “defiecient’

We thank the reviewer for noticing this typo. **We corrected it in the revised manuscript.**

Reviewer #2 (Remarks to the Author):

This manuscript by Kuhne et al. examines the regulation of neuronal morphogenesis by calcium, calmodulin, and the actin nucleation factor JMY. Using several immunoprecipitation strategies along with cell biological assays of dendritogenesis that combine RNAi of JMY with plasmid complementation using JMY mutants, the authors found that one of the actin-binding motifs (WH2#1) in the C-terminal region of JMY doubles as a calmodulin-binding sequence. This WH2#1 is crucial for JMY's to promote dendritogenesis. The Arp2/3 complex binding CA sequence at the C-terminus of JMY is also critical. Overall, this is an interesting paper packed with data. However, a few conclusions are incompletely supported by the images that were provided (points 1-3), and it is unclear if the authors have a model for how actin-, calmodulin-, and Arp2/3-binding are coordinated (points 4-6).

Specific Points

1. The endogenous localization of JMY in neurons in Figure 1E is unclear. Most JMY appears to be in the nucleus, and the so-called "dendritic localization pattern" of JMY and colocalization with MAP2 (lines 95-96) is weak and not very convincing. Do the authors have better images to support their conclusions that JMY is in dendrites? Do they have images of neurons stained for other WASP-family proteins to compare as negative controls?

We thank the reviewer for his/her critical assessment of the image shown in Fig. 1E. **We replaced the image by a more suitable representative example.** The new image more clearly shows a presence of JMY in the cytoplasm and also in the dendrites (highlighted by immunostaining against the dendritic marker MAP2).

The new anti-JMY immunolabeling furthermore clearly highlights the accumulation of JMY in dendritic growth cones (please see arrows in the revised Figure panel 1E), which unfortunately were not covered by the previous image. **The revised manuscript points out that also the JMY accumulation in dendritic growth cones in developing neurons points to some role of JMY in dendritic arbor formation.**

We hope that the reviewer will be content with the revised image and the more extensive description of the JMY localization in neurons in the revised manuscript.

Other WASP superfamily proteins, such as N-WASP, WAVE1 and WAVE3, are also expressed in neurons. They e.g. also show a presence and functions in the dendritic compartment of neurons during early development (e.g. see Banzai et al., 2000 *J. Biol Chem.*; Kessels & Qualmann, 2002 *EMBO J.*; Nozumi et al., 2003 *J Cell Sci.*). Both N-WASP and WAVE proteins also have been reported to be present in growth cones of different origin (e.g. see Kessels & Qualmann, 2002; Nozumi et al., 2003). The process of dendritic arbor formation and expansion thus seems to be rather complex and we are only beginning to identify and understand the molecular players and mechanisms involved.

2. Several of the measurements of neuronal morphogenesis are unclear to non-specialist readers. For example, in Figure 1 G,H,I,J, it would be very helpful if the authors provided magnified versions of individual neurons to explicitly highlight the number of dendritic branch/terminal points and length. These could be incorporated into the main figure or at the very least in the supplement. The methods in lines 695-697 should also be expanded beyond the citation of previous papers.

We thank the reviewer for the note that the quantitative analyses of neuronal development may need more explanations by further information/visualizations. **The revised manuscript now includes both more raw data information and more information on the evaluation:**

We included a schematic drawing of a developing neuron to illustrate the evaluated branch points, terminal points, dendritic branch depth and Sholl analyses (Supplementary Figure 1E).

We expanded Supplementary Figure 1D by adding the corresponding GFP channels of the neurons shown in Fig. 1G (GFP is used as a reporter for JMY expression; it also highlights the overall morphology of the transfected neuron but note that the endogenous anti-MAP2 immunostainings are used to track dendritic arbors of transfected cells) and enlarged these pictures, as the reviewer suggested.

In addition, we added pictures of the reconstruction of dendritic arborization by Imaris software (Imaris *filament*) to the revised Supplementary Figure 1D. We did the same for the example images shown in Fig. 2 (which furthermore also was complemented with the corresponding GFP channels). These Imaris reconstruction images also contain dendritic branch point and terminal point markings by Imaris (same color as in our scheme), which become visible when one zooms in. We added this information to the legends of Fig. S1 and Fig. 2.

We also expanded the description of the evaluations in the Material and Method section, as the reviewer suggested. It now refers to the schematic representation and the exemplary Imaris filaments we added to Figure S1 and Figure 2, which directly visualize the evaluations and thereby should help readers significantly to follow these detailed quantitative analyses of neuronal morphology.

3. Similar to point #2, in Figure 2A it is not apparent to the reader that the RNA#1-expressing cell in the middle of the image is different from any of the surrounding nontransfected cells.

We thank the reviewer for the hint that readers may have difficulties to see the reduced dendritic complexity caused by JMY deficiency. **We therefore have enlarged the images of the two independent JMY loss-of-function data sets (Fig. 2 and Fig. S2). In addition, we also added both the GFP reporter channel to both of these analyses (and to further figures in the manuscript) to mark the transfected cells better (previously just marked by an asterisk in the anti-MAP2 images) and a 2D representation of the 3D Imaris reconstruction of neuronal morphology (see revised Fig. 2A and revised Figure S2).** These Imaris reconstruction images also contain dendritic branch point and terminal point markings by Imaris (same color as in our scheme), which become visible when one zooms in. We added this information to the legends of Fig. S1 and Fig. 2. **We hope that the reduced dendritic complexity upon JMY RNAi can now be seen better in the revised figures.**

The nontransfected cells surrounding the transfected ones are not necessarily useful for analyses, as the morphometric analyses are based on the endogenous marker MAP2 and the to-be-evaluated neurons should thus only show minimal overlap with neighboring cells and this is often not given for all cells in a field of view. Therefore, separate control cells (scr. RNAi highlighted by GFP-reporter) have to be evaluated in all assays.

In general, we would like to note that the images accompanying the quantitative data sets of our study merely serve the purpose to illustrate the quantitative data. This also means they are chosen in a way that they are about representative of the means \pm SD of the quantitative data and therefore do not present phenotypes in an exaggerated way.

For the exact data, please also see our raw Data compilation added.

Moreover, in the accompanying Figure S2, it is unclear if is JMY depleted in the transfected cell, as MAP2 fluorescence also looks lower in the transfected cell. Can the authors estimate the % JMY knockdown by fluorescence intensity normalized to MAP2?

Fig. S2A illustrates the JMY knockdown visualized by anti-JMY immunostaining in developing neurons. It is correct that the image in Fig. S2A is only showing a moderate but still obvious difference between JMY RNAi#1 and the control cells – mostly because some weak signal remains in the JMY RNAi-

transfected cell. Quantitative assessments of the Fig. S2A image data yielded arbitrary fluorescence values of 53.5 for the JMY RNAi cell versus 78.4, 62.3, 71.0 and 90.1 for the surrounding control cells (i.e. JMY RNAi was 70.9% of mean control value in this image) and thereby confirmed the reduction of JMY levels seen in Fig. S2A (please also see **Reviewer Figure representing Fig. S2A with cell markings**).

Reviewer #2 Reviewer Figure 1. Version of Figure S2A illustrating control cells (numbers 1-4) evaluated for anti-JMY immunofluorescence signals in comparison to central JMY RNAi#1 cell (arrow). Arbitrary immunofluorescence values obtained from ROIs placed in the cell somata were 78.4, 62.3, 71.0 and 90.1 for the surrounding control cells (1-4) versus 53.5 for the JMY RNAi cell (arrow), i.e. 70.9% of mean of control cells. Bar, 10 μ m.

MAP2 immunostainings always show varying intensity levels in different cells irrespective of whether they are transfected or not and with which plasmid (see e.g. Fig. 1G; Fig. 2A, Fig. 3B or F, Fig. 4A or J and so on).

In order to characterize the JMY knockdown more extensively, we furthermore conducted a wider quantitative assessment of the JMY knockdown efficiency in developing neurons. The newly inserted quantitative data panel Fig. S2B shows that the JMY knockdown reaches at least a reduction towards a level of 70% and is of very high statistical significance (**, $p < 0.0001$) (revised Fig. S2B).** In addition to these clear knockdown results, it has to be taken into account that such quantitative immunofluorescence analyses of proteins with modest expression levels may lead to a gross underestimation of knockdown, as the technical background is somewhat difficult to specify.

Apart from these quantitative evaluations of JMY knockdown by anti-JMY immunostainings in neurons, the effectiveness and specificity of the JMY RNAi tools is furthermore shown by the fact that both JMY RNAi tools displayed clear and consistent loss-of-function phenotypes and that, most importantly, all dendritic parameters of both of these independent data sets were fully rescued by reexpression of RNAi-insensitive JMY.

Is the intensity or localization of F-actin or Arp2/3 complex staining altered in any way in JMY-depleted cells?

We have included some exemplary pictures into Fig. S4 of the revised manuscript. Gross changes in either F-actin or Arp2/3 complex localization upon JMY deficiency were not observed when JMY RNAi neurons compared to control neurons (please see revised Fig. S4C,D).

We moved the Sholl analyses of CK666-inhibited neurons (**former Fig. S4A**) into the main **Fig. 4 (new Fig. 4E)** in order to generate some space for the additional data the reviewer requested to be included in **Fig. S4**.

It would indeed have been interesting, if one could pinpoint when and where single actin cytoskeletal structures may be altered in JMY-deficient neurons. Similar to actin itself, however, the Arp2/3 complex is of course usually available in high excess in cells and the activation status of the complex cannot be seen directly by localization studies. Thus, overall anti-Arp2/3 complex immunostainings often look quite uninformative. Unfortunately, developing neurons are no exception from this rule. Rather on the contrary, as neurons also have no lamellipodia or similar structures with strong F-actin enrichments, also their overall F-actin distribution usually looks pretty uninformative.

The Figure 2 title stating that JMY is “required” for arborization is too strong, given that the most extreme phenotype in the figure (dendritic branch points) is only a ~2-fold reduction from the control.

In order to comply with the note of the reviewer, we have changed the title to “JMY is required for proper dendritic arborization” in the revised manuscript.

We would like to note that the observed dendritic branching phenotypes with JMY RNAi neurons having only about half the number of dendritic branch points belong to the strongest DIV4-to-DIV6 dendritic phenotypes reported in literature. The determined JMY loss-of-function defects easily reach e.g. the strong effects of the loss of the actin nucleator Cobl (Ahuja et al., 2007) and of its relative Cobl-like (Izadi et al., 2018).

Furthermore, it is commonly overlooked that already at DIV4, some dendritic arbor has developed, i.e. the starting point of dendritic branch formation under either control conditions or JMY RNAi is not zero.

4. Figure 6I seems like a key panel for examining the direct binding of calmodulin and actin to JMY, and for examining potential competition between calmodulin and actin. Can the authors gain more quantitative information about affinity by using different concentrations of the proteins?

The dissociation constants for G actin binding of WH2 domains usually are in the range of around 50 nM (MIM (Mattila et al. 2003); Cobl (Ahuja et al., 2007), WAVEs (Chereau et al., 2005)) to about 250 nM (WASP (Chereau et al., 2005); Spire (Sitar et al. 2011)) with some outliers as high as 2150 nM (WIP (Chereau et al., 2005)).

In relation to WH2 interactions, the CaM interaction with JMY is strong. We expected the K_D to be in the nM range similar to that of the tightly actin binding WH2 domains but in fact had to expand the concentration range examined multiple times to lower values in order to have a chance to resolve the tightness of CaM/JMY WH2#1 association in quantitative binding studies. **At 12.5 nM we unfortunately reached our detection limit. Thus, from our analyses, we can merely say that the K_D of the CaM binding to JMY is in the low nM range or even below. These data are now included in the revised Fig. S1 (Fig. S1C).**

The new data clearly highlight the fact that CaM binding to WH2#1 is strong enough to easily compete with the G-actin binding of WH2#1.

Perhaps more importantly, does calcium alone (in the absence of calmodulin) affect actin binding to JMY in these assays?

We experimentally addressed whether the actin binding to the JMY WH2 domain is modulated by the binding of the calcium sensor protein CaM or merely by the different calcium conditions applied. **The revised manuscript now covers these control experiments in detail. Figure S6E,F demonstrates by quantitative fluorescence-based Western blotting of *in vitro* reconstitutions of protein complex formations with immobilized GST-JMY-WH2#1 and actin that calcium itself has no effect on the actin binding of JMY’s WH2 domain #1.**

5. Similar to point #4, the conclusions of how calmodulin, actin, and Arp2/3 binding are coordinated are ambiguous. While point #4 can address the actin-calmodulin question, do the authors know whether WH2#1 combines with the CA sequence to promote Arp2/3 activation? The original results of Zuchero et al. imply that WH2#1-3 can nucleate actin directly, while WH2#3 can serve an Arp2/3-activating function with CA *in vitro*, but it is unclear if WH2#1 is functioning in direct nucleation or Arp2/3 activation in the current paper. What do the authors think is happening in their neuronal experimental system?

Pyrene-actin polymerisation assays indeed showed that a C-terminal fragment of JMY consisting only

of WH2#3 and the CA domain was sufficient for activating the Arp2/3 complex (Zuchero et al. 2009). Furthermore, JMY's three WH2 domains alone were sufficient for direct nucleation of actin (Zuchero et al. 2009). In line with these two separate functions, mutating only the first and second WH2 domain of JMY together blocked Arp2/3 complex-independent, i.e. JMY-mediated actin nucleation but still allowed activation of the Arp2/3 complex, as shown in *in vitro* pyrene actin assembly assays (Zuchero et al. 2012). One could therefore assume that these functions are fully segregated and the modulation of actin loading of the first WH2 domain by calmodulin is mostly affecting JMY's own actin nucleation capability. If one carefully goes through the published experiments, it becomes overt, however, that the highest actin nucleation activity was only obtained when the Arp2/3 complex was combined with all three WH2 domains and CA domain of JMY (Zuchero et al. 2012) – but this could of course also be due to additive effects of JMY's own nucleation capacity and that of its Arp2/3 complex activation.

Zuchero et al.'s mutational work *in vitro* was unable to dissect the two apparently interwoven mechanisms. Although maybe even more difficult to set up in a more complex *in vivo* system, such as developing neurons, by the use of CK666 we have provided evidence that Arp2/3 complex activation is needed for JMY's role in promoting dendritic branching.

Provided that this effect was directly related to the discovered Ca^{2+}/CaM regulation of the first WH2 domain, this may suggest an importance for the first WH2 domain in both, JMY's own actin nucleation capability and its capability to activate the Arp2/3 complex.

Alternative scenarios, however, could also be imagined. It would e.g. also be possible that JMY's intrinsic actin nucleation capability promotes the formation of some, first actin filaments (dependent on the first WH2 domain together with the second and third) and the Arp2/3 complex activation is then only needed for a subsequent process, which e.g. leads to more F-actin and then eventually powers dendritic branch induction or extension, which is the biological read-out. In this case the Arp2/3 complex would not be directly coupled to JMY but merely acting down-stream.

The revised manuscript now explains this better and thereby makes it clearer for the readers why it was important to dissect this by vigorously testing for a direct Arp2/3 complex-coupling of JMY by conducting rescue experiments with JMY^{*W981A} (Fig. 4).

Our studies demonstrated that JMY's critical role in dendritic branching does not solely rely on some indirect down-stream action of the Arp2/3 complex but does instead clearly require both the calmodulin-regulated, first WH2 domain being critical for JMY's own (*in vitro*) actin nucleation capability AND a direct Arp2/3 complex coupling of JMY via its Arp2/3 complex-binding and -activating C terminus.

We hope this becomes clearer in the revised manuscript.

6. The cartoon in Figure 8 summarizes the authors observations but does not address how the C-terminal functions of JMY are coordinated in the cell. Any advances related to points #4-5 would provide a clearer take-home message for the paper. Is calmodulin important for JMY localization or activation, and how/when might calcium fluctuations change the function of JMY? Insight (and some speculation) into the spatiotemporal regulation of JMY would be welcomed.

We thank the reviewer for this comment. **The revised manuscript now demonstrates that the overall localization of JMY upon CaM inhibition did not change (please see newly added Supplementary Figure S7).** Together with the fact that - at least in developing neurons - JMY is mostly localized in the cytoplasm of the cells (**revised Fig. 1E; see also localizations of JMY* and its mutants added throughout the manuscript during revision**), these data clearly suggest that the identified JMY functions in dendritic arbor formation direct represent actin cytoskeletal functions and their direct modulation by the calcium sensor protein CaM binding to the first WH2 domain of JMY.

The model in Fig. 8 summarizes this and shows how the discovered regulatory mechanisms by Ca^{2+}/CaM control the identified JMY function in promoting dendritic arborization.

The thus far not very well understood role of calcium transients in developing neurons is introduced and discussed in the manuscript. The manuscript also discusses the temporal aspects of the coordination of such signals with dendritic branch induction. JMY's cytoskeletal role in promoting dendritic branch induction would have to occur after calcium transients, as the calcium sensor protein CaM in its Ca^{2+} -

bound form would block the loading of JMY with G-actin. Thereby the prerequisite for JMY's actin cytoskeletal functions is under direct control of calcium transients sensed by CaM.

Minor

1. Lines 69-70, what makes this mechanism “powerful”?

We originally used the word to describe that CaM binding directly modulates the G-actin loading of JMY and therefore the most basic property, i.e. the prerequisite, for actin nucleation – a very elegant and very decisive point for regulation.

As the reviewer apparently was irritated by the use of the word “powerful”, **we changed the text of the revised manuscript to “an effective mechanism of regulation”**. We hope the reviewer will be more content with this.

2. Line 90 & 272, cell culture should not be classified as “in vivo” - a term which more often refers to experiments within a live animal.

The reviewer refers to mitotargeting (Line 90) in HEK293 cells and coimmunoprecipitations of endogenous actin from L929 cells (line 272), where we correctly stated that these experiments were conducted to address the *in vivo*-relevance of our findings.

We agree with the reviewer that “*in vivo*” has different meanings for different colleagues and that this can sometimes be confusing. While biochemists use *in vivo* for cellular extracts in order to tell such experiments apart from *in vitro* reconstitutions in conditions unrelated to life, and many cell biologists refer to *in vivo* when they study processes in living cells, other scientists, especially physiologists, use *in vivo* solely to refer to experiments in living model systems. **We have therefore changed the wording for the mitotargeting in cell cultures. The revised manuscript now states that the visual complex formation at the surface of mitochondria shows the JMY/CaM interaction “in intact cells”**.

The description of the ColPs did not state that these experiments were “*in vivo*” and could thus not be confused with the interpretation of the term by physiologists.

3. Line 113, what are Sholl analyses? Some contextualization for non-specialist readers (as in major point #2 above) would be appropriate here.

Please see our answer to the major point 2 above. Please also see the additional illustration added to Supplementary Figure S1 in the revised manuscript (Fig. S1C), which shows the concentric Sholl circles and should thereby directly illustrate why counting dendritic intersections with these circles provides a measure for dendritic complexity in different distances of the dendritic arbor from the cell body.

We also added the original reference to the revised manuscript (see Material and Method section as well as the Reference list: #50. Sholl, D.A. (1953) Dendritic organization in the neurons of the visual and motor cortices of the cat. *J Anat.* 87, 387-406.

4. The results in Figure 3 are not surprising, as (to my knowledge) no WASP-family member can be split in half and retain normal function in cells. If the paper needs focusing/shortening, this figure could be moved to the supplement.

We thank the reviewer for his/her suggestion. The revised Fig. 3 points out more clearly why the experiments shown in Fig. 3 are informative. We would like to refer the reviewer to the new schematic representation of the two mutants used in Fig. 3 for rescue attempts. The so-called CT mutant solely lacks the actin- and Arp2/3 complex-binding part of JMY but still contains all other domains including the p300 binding domain required for the proposed nuclear functions of JMY. The fact that it does not rescue thus clearly points out the JMY's effects in neuronal development are explicitly reflecting its actin

cytoskeletal functions. The second mutant then shows that, while actin and/or Arp2/3 binding is critical, it alone still is not sufficient for JMY's functions in dendritogenesis. **We hope that the revised manuscript explains this better.**

Apart from this, the reviewer may e. g. also find a recent Scar/WAVE paper from Buracco et al. (2024 *Current Biol.*) very interesting: "Here, we show in both B16-F1 mouse melanoma and *Dictyostelium discoideum* cells that Scar/WAVE without its VCA domain still induces the formation of morphologically normal, actin-rich protrusions, extending at comparable speeds despite a drastic reduction of Arp2/3 recruitment." (Buracco, S., Döring, H., Engelbart, S., Singh, S. P., Paschke, P., Whitelaw, J., Thomason, P. A., Paul, N. R., Tweedy, L., Lilla, S., McGarry, L., Corbyn, R., Claydon, S., Mietkowska, M., Machesky, L. M., Rottner, K., & Insall, R. H. (2024). Scar/WAVE drives actin protrusions independently of its VCA domain using proline-rich domains. *Curr. Biol.* 34(19), 4436–4451.e9.)

We added a sentence describing the observed mechanistical difference of JMY when compared to the recent and very interesting findings for Scar/WAVE by Buracco et al. 2024. **Please see Results section of the revised manuscript and Reference list (new Ref. #21), respectively.**

5. Lines 185-187, have any previous studies examined the impact of CK666 treatment on neuronal morphogenesis/dendritogenesis? If so, they should be mentioned/cited here.

We thank the reviewer for this suggestion. CK666-mediated Arp2/3 complex inhibition was reported to disrupt the localization of the Arp2/3 complex and to decrease F-actin barbed end density along the leading edge growth cones of *Aplysia* bag cell neurons (Yang et al. 2012). We added this information to the revised manuscript (new Ref.#24: Yang Q, Zhang XF, Pollard TD, Forscher P. 2012. Arp2/3 complex-dependent actin networks constrain myosin II function in driving retrograde actin flow. *J Cell Biol*, 197 (7):939-956).

6. Line 382, any in-text referral to unpublished observations should be removed or supported by data added to the supplement.

The reviewer refers to: "In contrast, our gain-of-function and particularly our rescue experiments in loss-of-function studies demonstrate that, in neurons but also in COS-7 cells, untagged full-length versions of JMY showed full activity in cellular morphogenesis. In contrast, N terminally tagged versions, such as GFP-JMY and Flag-JMY, did not (our unpublished observations)."

We have removed the latter sentence referring to our unpublished observations that we did not obtain successful rescues and also no gain-of-function phenotypes opposite to JMY loss-of-function phenotypes with tagged versions of JMY in developing neurons.

This is not of relevance for our study, as we exclusively worked with untagged JMY and JMY mutants in our functional analyses. The omitted information may, however, call for carefully revisits of JMY observation reported in the literature, as they are in part based on tagged JMY.

Reviewer #3 (Remarks to the Author):

This paper identified the WH2 domain-containing actin nucleator protein JMY as a novel, Ca²⁺-dependent direct binding partner of Calmodulin, and demonstrated that the complex of JMY and CaM regulates dendritic development. Through loss-of-function studies mediated by shRNA knockdown of JMY protein levels, coupled with overexpression of domain deletion constructs, the authors determined that JMY is necessary for normal dendritic branch development, and this function is dependent on both the N-terminal Arp2/3-binding domains and C-terminal G-actin-binding domain. Functional experiments further elucidated the domain in JMY that binds to CaM, which partially encompasses the first WH2 domain, and that JMY's first WH2 domain is the primary domain contributing in G-actin association. Finally, the association of JMY with actin through its first WH2 domain was shown to be regulated by Ca²⁺-activated CaM, with an increase in CaM binding resulting in a decrease in JMY actin binding. The following points need to be addressed.

Major points:

1. This study relies on the use of shRNA-mediated knockdown of JMY to determine the role of this protein in dendritic development. However, validation is needed to support the knockdown of JMY at the protein level. Fig. S2A shows immunofluorescence staining against JMY in a single neuron expressing RNAi#1. Given that the findings of this paper are dependent on knockdown of JMY, more validation is necessary for the shRNA knockdown. Ideally, this should be demonstrated by quantitative Western blotting against JMY in shControl or shJMY expressing neurons. Given the low transfection efficiency of primary neurons, immunofluorescence staining of neurons expressing control or JMY shRNA would also be acceptable, but a larger n than one neuron is necessary, and shControl-expressing neurons should also be included in this analysis.

Fig. S2A illustrates the JMY knockdown visualized by anti-JMY immunostaining in developing neurons. It is correct that the image in Fig. S2A is only showing a moderate but still obvious difference between JMY RNAi#1 and the control cells – mostly because some weak signal remains in the JMY RNAi-transfected cell. Quantitative assessments of the Fig. S2A image data yielded arbitrary fluorescence values of 53.5 for the JMY RNAi cell versus 78.4, 62.3, 71.0 and 90.1 for the surrounding control cells (i.e. JMY RNAi was 70.9% of mean control value in this image) and thereby confirmed the reduction of JMY levels seen in Fig. S2A (please also see **Reviewer Figure representing Fig. S2A with cell markings**).

Reviewer #2 Reviewer Figure 1. Version of Figure S2A illustrating control cells (numbers 1-4) evaluated for anti-JMY immunofluorescence signals in comparison to central JMY RNAi#1 cell (arrow). Arbitrary immunofluorescence values obtained from ROIs placed in the cell somata were 78.4, 62.3, 71.0 and 90.1 for the surrounding control cells (1-4) versus 53.5 for the JMY RNAi cell (arrow), i.e. 70.9% of mean of control cells. Bar, 10 μ m.

In order to characterize the JMY knockdown more extensively, we furthermore conducted a wider quantitative assessment of the JMY knockdown efficiency in developing neurons. The newly inserted quantitative data panel Fig. S2B shows that the JMY knockdown reaches at least a reduction towards a level of 70% and is of very high statistical significance (**, $p < 0.0001$) (revised Fig. S2B).** In addition to these clear knockdown results, it has to be taken into account that such quantitative immunofluorescence analyses of proteins with modest expression levels may lead to

a gross underestimation of knockdown, as the technical background is somewhat difficult to specify.

Apart from these quantitative evaluations of JMY knockdown by anti-JMY immunostainings in neurons, the effectiveness and specificity of the JMY RNAi tools is furthermore shown by the fact that both JMY RNAi tools displayed clear and consistent loss-of-function phenotypes and that, most importantly, all dendritic parameters of both of these independent data sets were fully rescued by reexpression of RNAi-insensitive JMY.

2. The neuronal images presented in this paper are shown with MAP2 staining as a marker of dendrites, and transfected cells are marked with a star. However, data showing multiple channel images with both GFP and MAP2 have not been included for any of the neuronal images, and should be included for transparency. This would allow comparison of transfected and non-transfected neighboring neurons.

We added the GFP channels for all the images either directly into the main figures or – due to space reasons – in the supplements.

In detail, the reviewer request applies to

Fig. 1G (corresponding GFP channel information added to **revised Fig. S1D**),

Fig. 2A (GFP images added to **revised Fig. 2A**),

Fig. S2C (GFP images added to **revised Fig. S2C**),

Fig. 3B (former Fig. 3A; GFP images added to **revised Fig. S3A**),

Fig. 4A (GFP images added to **revised Fig. 4A**),

Fig. S4A (GFP images added to revised Fig. S4 – **new Fig. S4A**),

Fig. S5A (GFP images added to **revised Fig. S5A**),

Fig. 6E (GFP images added to **revised Fig. S6B**),

Fig. 7A (GFP images added to **revised Fig. 7A**).

In addition, we added 2D representations of the Imaris-based 3D reconstructions of neuronal morphology to the data presented in both Fig. 1 (shown in an enlarged manner together with the GFP and anti-MAP2 images in the revised Fig. S1D) and in Fig. 2 (included together with the GFP images into the revised Fig. 2). These images clearly visualize the differences between the different tested experimental conditions.

The nontransfected cells surrounding the transfected ones are not necessarily useful for analyses, as the morphometric analyses are based on the endogenous marker MAP2 and the to-be-evaluated neurons should thus only show minimal overlap with neighboring cells and this is often not given for all cells in a field of view. Therefore, separate control cells (scr. RNAi highlighted by GFP-reporter) have to be evaluated in all assays.

In general, we would like to note that the images accompanying the quantitative data sets of our study merely serve the purpose to illustrate the quantitative data. This also means they are chosen in a way that they are about representative of the means \pm SD of the quantitative data and therefore do not present phenotypes in an exaggerated way.

For the exact data, please also see our raw Data compilation added.

3. In Figure 1C, CaM was immunoprecipitated from HeLa cell lysate, and JMY co-immunoprecipitation was observed only in the anti-CaM sample. However, there are bands present in the IgG control sample for CaM, making interpretation of this experiment challenging. A representative blot without bands in the IgG control lane for CaM would lend credence to the conclusions of this figure.

We replaced the blot image for the IP in Fig. 1C for an alternative one without the faint background traces seen in the IgG control lane (please see revised Fig. 1C).

4. Again in Figure 4, why not show the GFP-construct instead of indicating cells with an asterisk? This is important to show the localization and level of overexpression of the GFP-constructs

We thank the reviewer for this suggestion. See also point 2. **We were able to include the additional GFP images for all neuronal examples directly in the revised version of Fig. 4A.**

5. Please show the localization of the different GFP-tagged deletion constructs to show how different deletion domain constructs localize.

We agree with the reviewer that checking the localizations of JMY mutants in neurons is important and of course we did that, as **now pointed out in the Material and Method section of the revised manuscript. The revised manuscript now includes anti-JMY immunostainings to visualize the expression and localization of JMY* and the mutants thereof (please see additional figure panels Fig. 3F, Fig. 4J, Fig. 5D,K; Fig. S6D of the revised manuscript).** These images clearly show that re-expressed JMY* and all mutants thereof are available in excess (i.e. significantly above endogenous levels, which can almost not be seen at the exposure times chosen to visualizing the re-expressed JMY* and its mutants) and also showed no gross differences in expression level and localization when compared to wild-type JMY*.

These improvements in the revised manuscript should make it obvious for the reader that the reason for JMY* mutants failing to rescue the JMY loss-of-function phenotypes clearly was not their insufficient supply or stability but instead their dysfunctionality.

Minor points:

1. In figure 1D, the figure legend describes the scale bar as being 10 um in both images, but the cells in the lower image appear much larger than the cells in the top image. Representative images showing the same zoom for both conditions would strengthen the conclusion of this figure.

The cells were depicted at the same magnification. Following the suggestion of the reviewer, **we replaced the image for the Mito-mCherry control condition in Fig. 1D** with a picture of control cells that have sizes more similar to those of the cells with successful complex formation of more similar size (**please see revised Fig. 1D**).

2. JMY overexpression, either alone or as a rescue for JMY shRNA, is used in a number of figures in this paper. However, staining against JMY in these cells has not been performed. Inclusion of immunofluorescence staining of JMY would be informative about the degree of overexpression of JMY in these cells, especially in comparison to the surrounding cells with endogenous JMY expression.

The concern of the reviewer is justified. If mutants analyzed in rescue experiments would fail to reach at least the expression levels of endogenous protein, a not (fully) successful rescue could theoretically simply reflect a lack of the mutant rather than its hampered functionality. It therefore always needs to be checked that mutants analyzed *in vivo* express sufficiently well.

We did both, i) we analyzed by quantitative immunofluorescence analyses in developing neurons, whether the mutants express well above endogenous background and to somewhat equal levels (**see Reviewer Figure below**) and ii) furthermore evaluated their localizations. **All mutants - even the ones carrying larger deletions - are showing neuronal expression levels in the same order of magnitude as the wild-type JMY*.** Also when compared to each other, there were no statistical differences among the different JMY* mutants tested. All mutants thus clearly were reexpressed in excess over the endogenous JMY (**see Reviewer Fig. 1 below**).

We therefore decided to include images of all the mutants directly into the manuscript. The images we added to Fig. 3, Fig. 4, Fig. 5 and Fig. S6 (**see inserted panels Fig. 3F, Fig. 4J, Fig. 5D,K; Fig. S6D**) clearly make the points that **i) re-expressed JMY* and all mutants thereof are available in excess**

and that ii) **no gross differences are detected in neither expression levels nor localization when compared to wild-type JMY* reexpression** (in this context it may be noteworthy that the neurons expressing all mutants have far less elaborate dendritic trees. Thus, the JMY presence in the dendritic periphery may appear to be reduced in some cases but in fact JMY presence in the cellular periphery is simply reduced in accordance with the dendritic arbor).

Together these new evaluations clearly demonstrate that the reason for JMY* mutants failing to rescue clearly was not their insufficient supply or lack of stability but their dysfunctionality.

Reviewer #1 Reviewer Figure 1. JMY* and all of its mutants analyzed throughout the study are expressed and furthermore show a clear excess over endogenous JMY levels

Quantitative analyses of anti-JMY immunofluorescence signals of primary hippocampal neurons, that were transfected with JMY* and the indicated JMY* mutants. Data, mean±SEM visualized as bar/dot plots. Statistical analyses One-way ANOVA/Tukey's, not statistically different (n.s.) when compared to WT JMY* and when mutants were compared among each other.

Furthermore, as the reviewer suggested, the Material and Method section of the revised manuscript now points out that we of course checked the expression levels and localization of JMY* mutants.

Together, the improvements in the revised manuscript should make it obvious for the reader that the reason for JMY* mutants failing to rescue the JMY loss-of-function phenotypes clearly was not their insufficient supply or stability but clearly their dysfunctionality.

3. Minor grammatical or clarity issues were identified in the following lines:

a. Line 96: exiting is used rather than existing

We thank the reviewer for drawing our attention to this typo. **We corrected it.**

b. Line 208: yielded is misspelled

We thank the reviewer for drawing our attention to this typo. **We corrected it.**

c. Line 277: a word may be missing in this sentence after similarity: “The similarity _____ the actin coimmunoprecipitations levels by JMY CT and by JMY WH2#1 suggested that the actin association seen in coimmunoprecipitations mostly reflected the contribution of the first WH2 domain (Fig. 6C,D).”

We thank the reviewer for drawing our attention to this typo. **We corrected it by adding the missing word “of” to the revised sentence.**

d. Line 294: Controlled is misspelled

We thank the reviewer for drawing our attention to this typo. **We corrected it.**

e. Line 309: The underlined phrase is ambiguous: more than 80% of...? Clarification is needed here.

We rephrased the sentence in the revised manuscript and hope the reviewer agrees with us that the rephrased sentence now is easier to understand.